# MERIT: Maximum-normalized Element-wise Ratio for Language Model Large-batch Training

Yang Luo[1]  Zangwei Zheng[1]  Ziheng Qin[1]  Zirui Zhu[1]  Yong Liu[1]  Yang You[1]

## Abstract

Large-batch training has become a cornerstone in accelerating the training of deep neural networks, yet it poses challenges in optimization and generalization. Existing optimizers like AdamW present performance degradation during language models' large-batch training, due to the information bottleneck in attention layers caused by the sharp increase of max attention logit. While the LAMB optimizer partially addresses this issue, some attention layers still face this issue. The reason is that $l_2$-norm-based trust ratios in LAMB are less effective in directly influencing the max value of query/key weights. Furthermore, the weight-wise trust ratio in LAMB is error-prone as it overlooks relationships of weight values within rows or columns. Building on these observations, we propose a novel optimizer, MERIT, which leverages the max-norm to calculate the trust ratio to constrain the max attention logit more effectively. Moreover, we further construct element-wise trust ratios to provide more robust update scaling by focusing on local weight structures. Extensive experiments of large-batch training across various sizes of GPT-2 models demonstrate the superior performance of MERIT. Notably, during the training of GPT-2 Medium, MERIT enables a 6k batch size without any performance degradation compared to the standard batch size (480) with 48B training tokens. This work highlights the importance of considering the max attention logit and finer-granularity trust ratio in large-batch training. It successfully improves the training stability and paves the way for larger batch usage, enabling faster development and iteration of large language models. Code is available at https://github.com/NUS-HPC-AI-Lab/MERIT/.

[1]School of Computing, National University of Singapore, Singapore. Correspondence to: Yang You <youy@comp.nus.edu.sg>.

*Proceedings of the $42^{nd}$ International Conference on Machine Learning*, Vancouver, Canada. PMLR 267, 2025. Copyright 2025 by the author(s).

## 1. Introduction

The advent of large language models has revolutionized natural language processing, achieving unprecedented performance across a wide range of tasks (Touvron et al., 2023; Dubey et al., 2024; OpenAI, 2024). However, the increasing size and complexity of language models always result in a high time cost for the training. With the growing availability of powerful GPU clusters and specialized hardware accelerators, large-batch training can dramatically reduce the time required to train state-of-the-art models, making it possible to iterate faster and explore more ambitious architectures by processing more data in parallel.

While large-batch training offers the potential for increased parallelism and faster convergence, it also introduces complex optimization dynamics that can impede model performance and stability (Keskar et al., 2017; Goyal et al., 2018; Shallue et al., 2019). Training large language models with large batches typically encounters one main issue: research has shown that training with large batches often leads to models performing poorly on unseen data. When using AdamW optimizer with large batch size, Figure 1 shows clear performance degradation, requiring additional training tokens to reach comparable generalization levels.

This paper identifies a crucial problem in large-batch training of language models: we observe the sharp increase of max attention logit in attention layers during the training process using AdamW optimizer (Kingma & Ba, 2017; Loshchilov & Hutter, 2019). The inflated max attention logit can result in overly sharp attention distributions, potentially causing the model to focus on specific tokens or patterns overly, thus hindering its ability to capture overall information in the data (Zhai et al., 2023).

While LAMB (You et al., 2020) successfully reduces the max attention logit in the first layer of GPT-2 models (Radford et al., 2019; Brown et al., 2020) by applying a weight-wise trust ratio, it faces limitations in further decreasing the value in the medium layer. The limitation arises from the lower efficiency of $l_2$ norm than max norm (largest absolute value) in preventing query/key weights from reaching extremely high values.

Moreover, our analysis reveals that rows and columns in

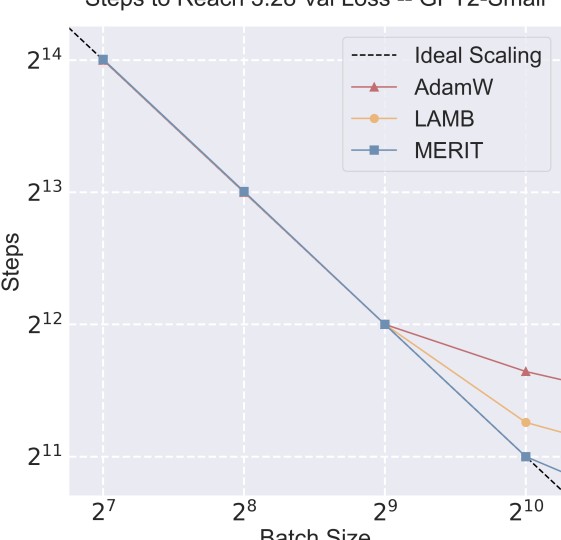
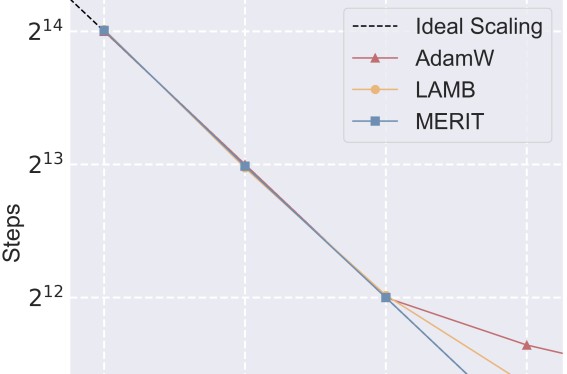

*Figure 1.* For both GPT-2 models, the connection between batch size and the number of steps required to reach a specific validation loss follows a similar pattern. At first, as the batch size increases, there is a phase of ideal scaling (shown by a dotted line) where doubling the batch size cuts the necessary steps in half. This is followed by a period where the benefits start to decrease. Eventually, a point is reached where further increasing the batch size (data parallelism) offers no additional advantage. This final stage represents the upper limit of large-batch training effectiveness.

large-batch trained weights often share similarities. The neglect of these relationships in the weight-wise ratio method proposed in LAMB leads to training instability as it fails to mitigate the negative impact of extreme values from other rows or columns. Given rows and columns exhibit high similarity, it allows for calculating element-wise ratios by considering the weights within the same rows/columns, while eliminating influences from other rows/columns. The proposed finer-granularity ratios focus on local weight structures, resulting in more stable large-batch training for language models.

Inspired by these insights, we propose a novel optimizer, MERIT, introducing max-norm-based trust ratios to precisely limit the maximum of query/key matrix and finer-grained ratios for focusing on specific weight structures. We conduct extensive experiments to evaluate the performance of MERIT compared to existing optimizers across various sizes of GPT-2 models. Our findings demonstrate the potential of MERIT to enhance large-batch training by improving convergence properties and generalization performance. This work contributes to the ongoing exploration of optimization strategies in large-batch training and highlights the significance of finer-grained ratio calculation in designing effective optimizers.

## 2. Related Work

### 2.1. Large-batch Training

Scaling up batch sizes during the training of deep neural networks has been an active area of research, as it allows for better parallelization across multiple GPUs and reduces time-to-train. However, naively increasing the batch size often leads to degraded model performance, a phenomenon dubbed the "generalization gap". Several techniques have been proposed to enable large-batch training without compromising accuracy. Goyal et al. (2018) showed that linear scaling of the learning rate with respect to the batch size can maintain model quality for batch sizes up to 8K on ImageNet. Other works proposed novel optimization algorithms like LARS (You et al., 2017) and LAMB (You et al., 2020) that dynamically adapt layer-wise learning rates based on parameter norms and momentum. Liu et al. (2022) introduced a more efficient SAM (Foret et al., 2021) variant for training Vision Transformers (Dosovitskiy et al., 2021) using large batches and Luo et al. (2023) explored memory-efficient optimization techniques for large-batch training of language models. However, LAMB still presents a large max attention logit and shows a weight-wise trust ratio calculation containing some error, leading to large-batch training performance degradation.

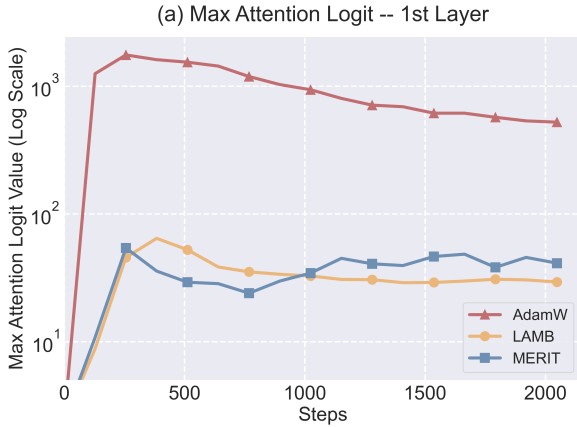

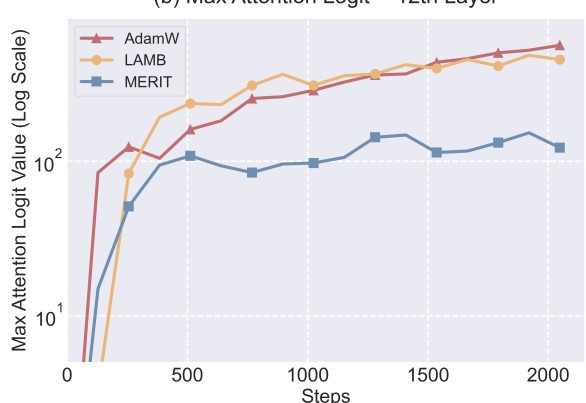

*Figure 2.* Max attention logit of self-attention layers during the large-batch training of GPT-2 medium model using three optimizers. (a) Max Attention Logit of first self-attention layer. (b) Max Attention Logit of medium (12th) self-attention layer. A comprehensive visualization is availabe in Appendix I.

## 2.2. Max Attention Logit

The relationship between max attention logit and training stability of transformers has been explored extensively. Researchers have previously documented that Transformer training fails when the attention logits become large. Dehghani et al. (2023) addressed the challenge of uncontrolled growth in attention logits in large-scale transformers by implementing query/key normalization, effectively stabilizing the training process and preventing the near one-hot attention distributions typical in models with parameters nearing 8 billion. Wortsman et al. (2024) observed the loss diverges and training fails when the max attention logit exceeds approximately $10^4$. Zhai et al. (2023) identified attention entropy collapse as a common issue in Transformer training across various domains and tasks and proposed $\sigma$Reparam to reparameterize the weights of linear layers using spectral normalization and a learned scalar. Nevertheless, the significant increase in the max attention logit value during large-batch training remains unexplored and leads to the poor performance of existing optimizers.

## 3. Preliminaries

### 3.1. Max Attention Logit Growth in Large-batch Training

In the self-attention layer of a Transformer (Vaswani et al., 2017), attention logits are calculated by combining queries $Q_i$ and keys $K_i$ using the formula $z_{ij} = \langle Q_i, K_j \rangle / \sqrt{d_k}$, where $d_k$ represents the head dimension. These logits are then processed through a softmax function to generate attention weights to aggregate values $V_i$. The max attention logit is defined as the max value among the computed attention logits, $\max z_{ij}$. Dehghani et al. (2023) observed that the attention logits $z$ became large when using relatively

high learning rates, which they termed as attention logit growth. Consequently, the attention weights collapse to one-hot vectors and cause unstable training, a phenomenon termed attention entropy collapse by Zhai et al. (2023).

In the large-batch training of GPT models, larger learning rates are needed compared to normal batch sizes, AdamW-based training consistently presents a similar max attention logit sharp increase that leads to one-hot-vector attention output as analyzed in Appendix C, limiting the expression ability of attention layers. As shown in Figure 2(a), the max attention logit of the first self-attention layer during large-batch training with AdamW significantly exceeds the value observed in small-batch training presented in Figure 11, leading to training instability and performance degradation. As a result, AdamW-based large-batching leads to a worse generalization performance than small batches.

### 3.2. Trust ratio in LAMB

A distinguishing feature of the LAMB optimizer is its implementation of the "trust ratio", a mechanism designed to dynamically adjust the learning rates for each neural network layer based on their respective weight norms and update norms. The trust ratio $R$ for particular weights $w$ at time $t$ is defined as the ratio of the $l_2$-norm of weights to that of updates $u$:

$$R = \frac{\|w_t\|}{\|u_t + \lambda w_t\|} \tag{1}$$

where $u_t = \frac{m_t}{\sqrt{v_t} + \epsilon}$, $\|\cdot\|$ denotes $l_2$-norm and $\lambda$ is the weight decay parameter. Since large-batch training employs high learning rates, LAMB addresses the issue of excessively large or small gradients/updates by adaptively rescaling LRs per layer. This ensures that updates remain aligned with the

scale of the initial weights.

Through the design of trust ratios, LAMB optimizer achieves a balance that allows it to exploit the computational benefits of large batch sizes without compromising the robustness of the model training process. However, the increment of max attention logit still exists during large-batch training as presented in Figure 2(b).

# 4. Algorithm

## 4.1. Maximum Normalized Ratio

The max attention logit is directly relevant to the max norm (largest absolute value) in key matrix $W_K$ and query matrix $W_Q$, as evidenced by the equation: attention logits $z = XW_KW_Q^\top X^\top / \sqrt{d_k}$, where $X$ represents the input sequence to a self-attention layer. Hence, the issue outlined in Section 3.1 can be addressed by considering max norm when developing the trust ratio for training with large batches. A complete analysis is presented in Appendix D.

However, there is a huge disparity between the max norms of query and key weights and their corresponding $l_2$ norms (Appendix E). In this case, $l_2$-norm-based trust ratios cannot effectively prevent the huge increase of the maximum of $W_Q/W_K$. Thus, LAMB often fails to reduce max attention logit in the medium self-attention layer further as depicted by Figure 2(b). We modify the LAMB optimizer using the max norm instead of the $l_2$ norm when calculating the trust ratio. Therefore, the proposed method gives larger updates to extreme values of weights. This helps prevent extreme values in query and key weights from becoming too large, limiting spikes in the maximum attention logit.

## 4.2. Element-wise Trust Ratio

To further improve the large-batch training performance of layer-wise ratios, we devise an element-wise ratio to capture local weight structures more accurately while maintaining computational efficiency. Due to the multi-headed self-attention mechanism and outlier dimension phenomenon observed throughout the training process of transformers (Kovaleva et al., 2021; Puccetti et al., 2022), weight values exhibit similarities within rows/columns as shown in Figure 3. Given this context, weight-wise trust ratios of LAMB will introduce certain inaccuracies because extreme values in one row/column can adversely impact the training stability of other rows/columns.

To address this limitation, we propose a novel approach that employs an element-wise ratio to leverage the inherent similarity of weights within the same rows or columns of the weight matrix. Our method involves calculating ratios along both rows and columns and then selecting the larger of these two values for each element.

Specifically, let $w \in \mathbb{R}^{n \times n}$ and $u \in \mathbb{R}^{n \times n}$ be the weight matrix and update matrix separately, with $w^{(i)}$ representing elements at the $i$-th row and $w^{(j)}$ representing elements at the $j$-th column. We first calculate the row-wise ratio $r$ for each row $r^{(i)} = \|w^{(i)}\|_m / \|u^{(i)}\|_m$ and the column-wise ratio $c$ for each column $c^{(j)} = \|w^{(j)}\|_m / \|u^{(j)}\|_m$. Lastly, the final element-wise ratio $s$ is determined by $s^{(i,j)} = \max\{r^{(i)}, c^{(j)}\}$. This improved approach seeks to boost the capacity of the optimizer to adjust to specific weight structures in different parts of the network, leading to improved convergence and generalization performance in language models.

## 4.3. MERIT

Algorithm 1 summarizes our proposed MERIT optimizer. The design of MERIT comes from two parts: maximum-normalized trust ratio and element-wise refinement. Finally, we implement an element-wise clipping mechanism that limits the max update magnitude to 1 across all parameter dimensions, which mirrors the update strategy of stochastic Sign Momentum Gradient Descent (Bernstein et al., 2018), serving to enhance the overall stability of the large-batch optimization process. The designs incorporated in MERIT successfully address the issue of rapidly increasing max attention logits in the middle self-attention layers of language models, as illustrated in Figure 2(b).

---

**Algorithm 1** MERIT

1: **Input:** $x_1 \in \mathbb{R}^d$, learning rate $\{\eta_t\}_{t=1}^T$, parameters $0 < \beta_1, \beta_2 < 1, \epsilon > 0, m_0 = 0, v_0 = 0$
2: **for** $t = 1$ **to** $T$ **do**
3:      $g_t = \frac{1}{|X_t|} \sum_{x_t \in X_t} \nabla \ell(w_t, x_t)$.
4:      $m_t \longleftarrow \beta_1 m_{t-1} + (1 - \beta_1) g_t$
5:      $v_t \longleftarrow \beta_2 v_{t-1} + (1 - \beta_2) g_t^2$
6:      $u_t = \frac{m_t}{\sqrt{v_t} + \epsilon}$
7:      Weight-wise Ratio $\boldsymbol{b}_t = \frac{\|w_t\|_m}{\|u_t + \lambda w_t\|_m}$
8:      Row-wise Ratio $\boldsymbol{r}_t^{(i)} = \frac{\|w_t^{(i)}\|_m}{\|u_t^{(i)} + \lambda w_t^{(i)}\|_m}$
9:      Column-wise Ratio $\boldsymbol{c}_t^{(j)} = \frac{\|w_t^{(j)}\|_m}{\|u_t^{(j)} + \lambda w_t^{(j)}\|_m}$
10:     Element-wise Ratio
11:     $\boldsymbol{s}_t^{(i,j)} = \max\{\max\{\boldsymbol{r}_t^{(i)}, \boldsymbol{c}_t^{(j)}\}, \boldsymbol{b}_t\}$
12:     $w_{t+1} = w_t - \eta_t \cdot \mathbf{clip}(\boldsymbol{s}_t \cdot (u_t + \lambda w_t), 1)$
13: **end for**

---

## 4.4. Convergence Analysis

**Notation.** Let $\mathbb{I}$ be the $d \times d$ identity matrix, and let $\mathbb{I} = [\mathbb{I}_1, \mathbb{I}_2, ..., \mathbb{I}_h]$ be its decomposition into column submatrices $\mathbb{I}_i = d \times d_h$. For $w \in \mathbb{R}^d$, let $w^{(i)}$ be the block of variables corresponding to the columns of $I_i$ i.e., $w^{(i)} = \mathbb{I}_i^T w \in \mathbb{R}^{d_i}$ for $i = 1, 2, \cdots, h$. For any function $f : \mathbb{R}^d \to \mathbb{R}$, $\nabla_i f(w)$ denotes the gradient with respect to

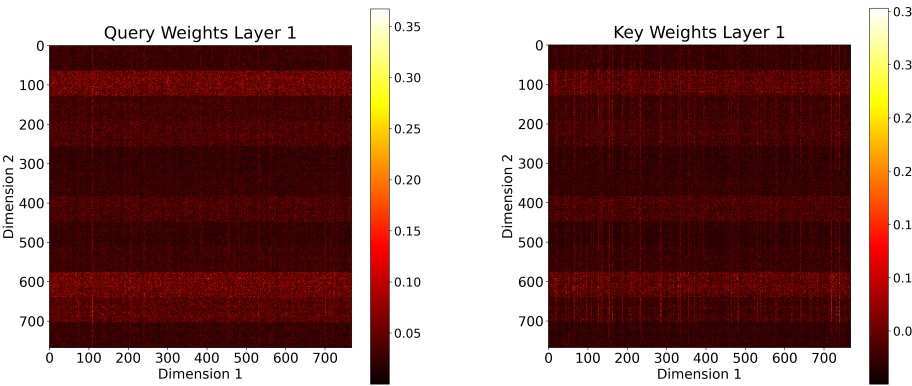

*Figure 3.* Analysis of GPT-2 small's first attention layer reveals patterns in query-key weight magnitudes: weights show high similarities within both rows (arising from multi-headed attention architecture) and columns (due to the outlier dimension phenomenon).

$w^{(i)}$. For vectors $u$ and $v \in \mathbb{R}^d$, we use $u^2$ to represent the element-wise square operation and $u/v$ to represent the element-wise division operation. We use $\| \cdot \|$, $\| \cdot \|_1$ and $\| \cdot \|_m$ to denote $l_2$-norm, $l_1$-norm and max-norm of a vector respectively. Consider the following nonconvex stochastic optimization problems of the form

$$\min_{w \in \mathbb{R}^d} f(w) := \mathbb{E}_{x \sim \mathbb{P}}[\ell(w, x)] + \frac{\lambda}{2} \|w\|^2, \qquad (2)$$

where $w$ is model parameters to optimize, $\ell$ is the loss function and $\mathbb{P}$ is a probability distribution on the unknown training data $\mathcal{X} \subset \mathbb{R}^k$.

**Assumption 1.** The loss function $\ell(u)$ is $L_i$-smooth with respect to $u^{(i)}$, which means there exists a non-negative constant $L_i$, $\forall u, v \in \mathbb{R}^d$, and $x \in \mathcal{X}$:

$$|\nabla_i \ell(u, x) - \nabla_i \ell(v, x)| \le L_i |u^{(i)} - v^{(i)}|, \qquad (3)$$

for all $i \in [h]$. Let $L = (L_1, \cdots, L_h)^T$ represent the vector of Lipschitz constants in $h$ dimensions. We use $L_{avg}$ to denote $\sum_i \frac{L_i}{h}$.

**Assumption 2.** The variance in stochastic gradients is subject to the following upper bound:

$$\mathbb{E}|\nabla_i \ell(w, x) - \nabla_i f(w)|^2 \le \sigma_i^2 \text{ for all } w \in \mathbb{R}^d \text{ and } i \in [h]$$
$$\mathbb{E}|[\nabla \ell(w, x)]_i - [\nabla f(w)]_i|^2 \le \tilde{\sigma}_i^2 \text{ for all } w \in \mathbb{R}^d \text{ and } i \in [d], \qquad (4)$$

and $\sigma = (\sigma_1, \cdots, \sigma_h)^T$ and $\tilde{\sigma} = (\tilde{\sigma}_1, \cdots, \tilde{\sigma}_d)^T$ are used to denote the vectors of standard deviations of stochastic gradient per layer and per dimension separately.

**Assumption 3.** Gradients are bounded i.e., $[\nabla l(w, x)]_i \le G$ for all $i \in [d]$, $w \in \mathbb{R}^d$ and $x \in \mathcal{X}$. Note that such assumptions are typical in the analysis of stochastic first-order methods.

Due to the usage of element-wise clipping on controlling the worst-case (largest) update size in all parameter dimensions

to be at most 1, the training stability is improved and we only need to consider the convergence analysis of weight-wise maximum-normalized ratio that is the lower bound of the proposed MERIT, which is noted as MERIT-W. The following result provides a convergence rate for MERIT-W in general nonconvex settings. Following the analysis in (You et al., 2020), we focus on the setting where $\beta_1 = 0$ and $\lambda = 0$.

**Theorem 1.** Let $\eta_t = \eta = \sqrt{\frac{2(f(w_1) - f(w^*))}{\alpha_u^2 \|L\|_1 dT}}$ for all $t \in [T]$, $b = T$, $d_i = d/h$ for all $i \in [h]$, and $\alpha_l \le \|v\|_m \le \alpha_u$ for all $v > 0$ where $\alpha_l, \alpha_u > 0$. Then for $w_t$ optimized by MERIT-W, we have the following bounds:

1. When $\beta_2 = 0$, we have

$$\left( \mathbb{E} \left[ \frac{1}{\sqrt{2 \log(d)}} \|\nabla f(w_a)\|_1 \right] \right)^2 \le$$
$$O \left( \frac{(f(w_1) - f(w^*)) L_{avg}}{T} + \frac{\|\tilde{\sigma}\|_1^2}{T dh} \right),$$

2. When $\beta_2 > 0$, we have

$$\mathbb{E}[\|\nabla f(w_a)\|^2] \le$$
$$O \left( \sqrt{\frac{2G^2 \log(d)}{h(1 - \beta_2)}} \times \right.$$
$$\left. \left[ \sqrt{\frac{2(f(w_1) - f(w^*))\|L\|_1}{T}} + \frac{\|\tilde{\sigma}\|_1}{\sqrt{T d}} \right] \right),$$

where $w^*$ represents an optimal solution to the problem outlined in equation 2 and $w_a$ is an iteration uniformly randomly selected from $\{w_1, \cdots, w_T\}$. For a detailed proof of convergence, please refer to Appendix J.

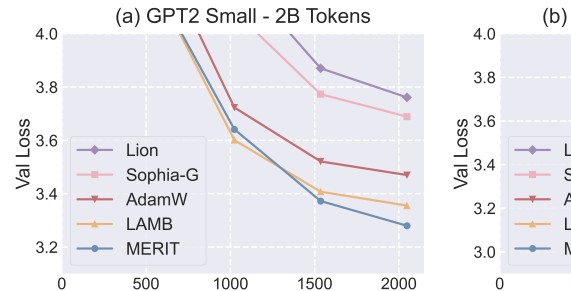 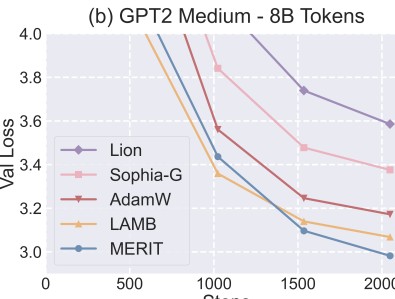 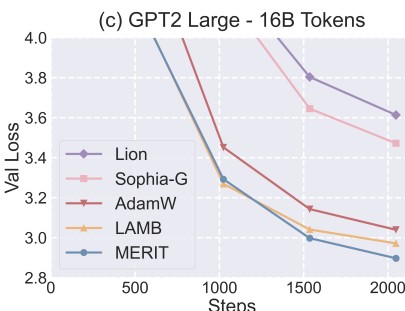

*Figure 4.* Final validation loss. (a) GPT-2 Small (125M, batch size=1K). AdamW: 3.470, LAMB: 3.355, MERIT: 3.280 (b) GPT-2 Medium (355M, batch size=4K). AdamW: 3.172, LAMB: 3.068, MERIT: 2.982. (c) GPT-2 Large (770M, batch size=8K). AdamW: 3.039, LAMB: 2.971, MERIT: 2.897.

## 5. Experiments

### 5.1. Setup

**Language modeling.** We conducted large-batch training experiments on OpenWebText (Gokaslan & Cohen, 2019), training autoregressive models from scratch using settings derived from the Chinchilla scaling law (Hoffmann et al., 2022). Following standard protocol, we set the context length of GPT-2 to 1024. Our experiments encompassed three model sizes: 125M (small), 355M (medium), and 770M (large). Detailed specifications of the model configurations can be found in Appendix A.

**Baselines.** We compare MERIT with LAMB, the dominantly used optimizer on large-batch training of language modeling tasks, Adam with decoupled weight decay (AdamW), Lion (Chen et al., 2023), and Sophia-G (Liu et al., 2024). For all models, all learning rates are tuned with grid search. The weight decay is set to 0.1 for all optimizers for a fair comparison. We follow Liu et al. (2024) for the settings of $\beta$ values: For AdamW: $\beta_1 = 0.9$ and $\beta_2 = 0.95$. For Lion: $\beta_1 = 0.95$ and $\beta_2 = 0.98$. For Sophia-G: $\beta_1 = 0.92$ and $\beta_2 = 0.99$.

**Chinchilla Scaling Law.** The Chinchilla scaling law suggests an optimal training regime of approximately 20 training tokens per parameter for large language models (Anil et al., 2023; Muennighoff et al., 2023). This principle, derived from Hoffmann et al. (2022)'s comprehensive analysis, proposes that model performance is maximized when the number of training tokens scales proportionally with the number of parameters under a fixed compute budget. Following this established benchmark allows for a fair comparison with other research in the field with limited computing resources.

**Implementation.** Following the Chinchilla scaling law, we use batch size 1K for GPT-2 small with 2B training tokens, 4K for GPT-2 medium with 8B tokens, and 8K for GPT-2 large with 16B tokens for the large-batch training setting.

Our learning rate (LR) follows a cosine schedule, with the final LR set to 0.1 of the peak LR. We maintain a constant LR warm-up ratio of 0.02 and apply standard gradient clipping (norm) with a threshold of 1.0. In the case of Sophia-G, we select 240 examples from each minibatch to compute the diagonal Gauss-Newton and update the diagonal Hessian every 10 steps. We implement the algorithms in PyTorch (Paszke et al., 2019) and train all the models in bfloat16. All models are trained on H100 GPUs.

**Technical details.** We mainly evaluate GPT-2 models with their log perplexity and plot the validation loss curves. The results from LAMBADA (Paperno et al., 2016), WikiText (Merity et al., 2017), and SuperGLUE (Wang et al., 2019) evaluations are also included in our experiments.

### 5.2. Results

Figure 4 illustrates the validation loss curve on OpenWebText with the same number of steps. MERIT consistently achieves lower validation loss than LAMB, AdamW, Lion, and Sophia-G. As the size of the language model increases, the performance gap between MERIT and baselines becomes larger. Besides, during large-batch training of GPT-2 Large, the performance gap between AdamW and LAMB diminishes. In contrast, MERIT demonstrates an increased advantage over LAMB under this condition. MERIT achieves a 0.07 lower validation loss on the 123M model (Figure 4 (a)) with the same training tokens, which means a significant improvement according to training scaling laws in this regime (Kaplan et al., 2020; Hoffmann et al., 2022; Liu et al., 2024).

**The scaling law favors MERIT over LAMB.** Figure 1 illustrates the number of steps required for GPT-2 models with varying batch sizes to reach equivalent validation loss on OpenWebText. The figure reveals a noticeable decline in generalization performance when training language models with large batch sizes using AdamW. Importantly, MERIT enables the use of larger batch sizes without compromis-

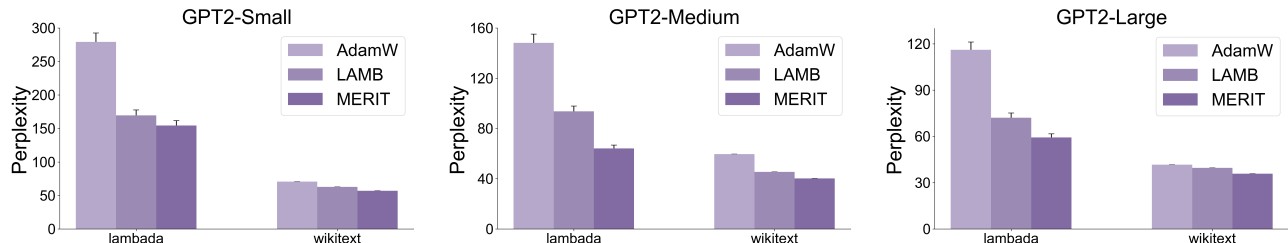

*Figure 5.* Zero-shot evaluation on LAMBADA and WikiText. With the same number of steps, language models large-batch pre-trained with MERIT outperform models pre-trained with AdamW and LAMB on both tasks with lower perplexity scores.

*Table 1.* Comparison of zero-evaluation performance for GPT-2 Small and GPT-2 Medium models under Sophia training settings.

| Model | ARC ↑ | COPA ↑ | HelllaSwag ↑ | RACE ↑ | WIC ↑ | Avg ↑ |
|---|---|---|---|---|---|---|
| GPT-2 Small (AdamW-Batch Size=**480**) | 43.43 | 66.00 | 29.20 | 29.00 | 50.16 | 43.56 |
| GPT-2 Small (MERIT-Batch Size=**4k**) | 45.83 | 67.00 | 28.82 | 27.56 | 50.16 | **43.87** |
| GPT-2 Medium (AdamW-Batch Size=**480**) | 49.49 | 71.00 | 32.39 | 30.05 | 50.00 | 46.59 |
| GPT-2 Medium (MERIT-Batch Size=**6k**) | 50.38 | 70.00 | 32.32 | 30.33 | 50.47 | **46.70** |

ing performance (1K for GPT-2 small and 4K for GPT-2 medium) than LAMB. Moreover, the performance gap between MERIT and LAMB, given the same number of training tokens, widens for 355M parameter models compared to 125M parameter models.

**Zero-shot Evaluation.** The enhanced validation loss performance translates to better results in evaluation tasks as shown in Figure 5. We measure the zero-evaluation performance of trained GPT models on LAMBADA and WikiText using perplexity scores. MERIT successfully obtains lower perplexity across both tasks compared with AdamW and LAMB. Our evaluation focuses solely on zero-shot performance for pre-trained GPT-2 models, as demonstrating in-context learning typically requires GPT models with at least a billion parameters. Additional evaluation results are available in Appendix G.

**Training Results on Llama.** We conducted additional experiments using C-Optim [1] to validate the performance of the proposed MERIT optimizer in the large-batch training of Llama models (Dubey et al., 2024). In alignment with Chinchilla scaling laws, we trained models on 2.6B tokens from the C4 dataset (Raffel et al., 2023) (batch size=1K) for GPT-2 small and 8B tokens (batch size=4K) for GPT-2 medium. We maintained the same hyperparameter settings, specifically a weight decay of 0.1 and beta values of 0.9 and 0.95. Figure 6 demonstrates that MERIT consistently improves performance across different language model architectures in large-batch training scenarios.

---
[1]https://github.com/kyleliang919/C-Optim

### 5.3. Further Analysis

**Performance Gap Between Standard Batch Size and Large Batch Size.** We conduct experiments following the training protocol outlined in Liu et al. (2024), using 48 billion tokens for training. Our study compares MERIT's performance with large batches against AdamW's performance with small batches (batch size=480). As illustrated in Table 1, MERIT demonstrates the ability to increase batch sizes to 4K for GPT-2 small pre-training and 6K for GPT-2 medium pre-training without compromising generalization performance. These findings suggest that MERIT enables language models to utilize larger batch sizes as the model scale increases.

**Curvature of Convergence Point.** The curvature of the convergence point in the loss landscape differs significantly between small and large batch sizes, impacting model generalization and robustness. Large batch sizes often lead to convergence in sharper minima with higher curvature (Keskar et al., 2017). While these sharp minima may achieve lower training loss, they can result in poorer generalization due to their sensitivity to small changes.

In Figure 7(a), we present the eigenvalue distributions of Hessian matrices at the convergence points of GPT-2 small models pre-trained using AdamW and MERIT algorithms. The convergence point achieved by MERIT exhibits a smaller top eigenvalue (12.326) and trace (3444.92) than AdamW whose top eigenvalue and trace equal 37.231 and 12994.91 respectively, and eigenvalues of MERIT are predominantly confined to the range $[-5, 5]$. This reduced spread of eigenvalues suggests that MERIT converges to an overall flatter region in the optimization landscape. Such

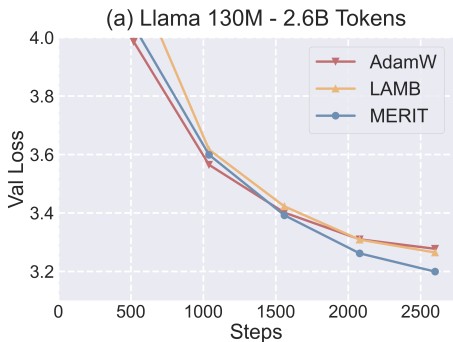
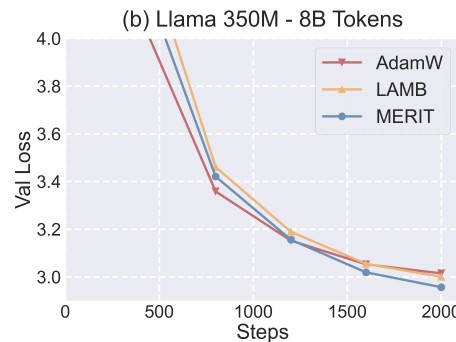

*Figure 6.* Final validation loss. (a) Llama 130M (batch size=1K). AdamW: 3.277, LAMB: 3.265, MERIT: 3.199 (b) Llama 350M (batch size=4K). AdamW: 3.014, LAMB: 3.001, MERIT: 2.957.

flat regions, characterized by small eigenvalues and trace, are frequently associated with improved generalization capabilities in language models.

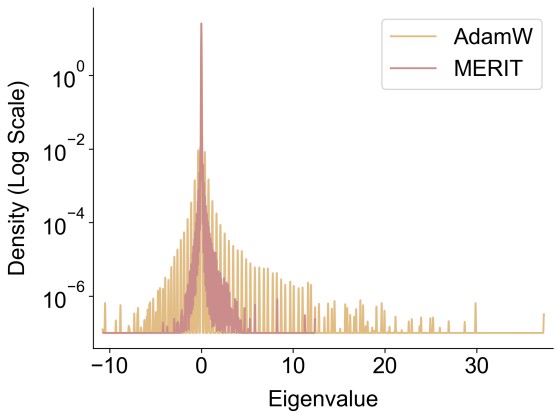

*Figure 7.* A graphical representation comparing the eigenvalues of Hessian matrices at convergence points, contrasting models pre-trained using AdamW versus MERIT..

**Comparison of wall-clock time and computational resources.** In Table 2, we present a comparison of the total computational requirements (measured in TFLOPS) per step and the actual time taken (wall-clock time) on A100 GPUs. Following the methodology of Chowdhery et al. (2022), we report the average time per step (T(step)) and corresponding FLOPS. The data in Table 2 reveals that employing maximum-normalized and element-wise trust ratio calculation adds minimal extra computational overhead (1%) compared to $l_2$-norm-based trust ratio of LAMB. Overall, the increase in FLOPS is negligible compared to AdamW and LAMB.

**QK-Norm VS MERIT.** QK-Norm (Dehghani et al., 2023) was developed to mitigate training instabilities encountered when scaling Vision Transformer (ViT) models to unprecedented sizes with higher learning rates. This technique applies Layer Normalization to the query and key vectors prior

*Table 2.* Wall-clock time and TFLOPS.

| Optimizer | Model Size | T(step) | TFLOPS |
|-----------|-----------|---------|--------|
| AdamW | 770M | 242.50s | 43.91 |
| LAMB | 770M | 243.51s | 43.73 |
| MERIT | 770M | 245.46s | 43.38 |
| AdamW | 355M | 57.69s | 44.05 |
| LAMB | 355M | 57.93s | 43.86 |
| MERIT | 355M | 58.50s | 43.43 |

to the attention computation in the transformer architecture. However, as illustrated in Figure 8(b), while QK-Norm enables larger learning rates for AdamW in GPT-2 models, it negatively leads to performance degradation in large-batch training. A potential explanation for this discrepancy is that QK-Norm aims to stabilize attention computations and inadvertently restricts information flow within the attention layers of language models, which becomes more obvious in large-batch training with much fewer optimization steps.

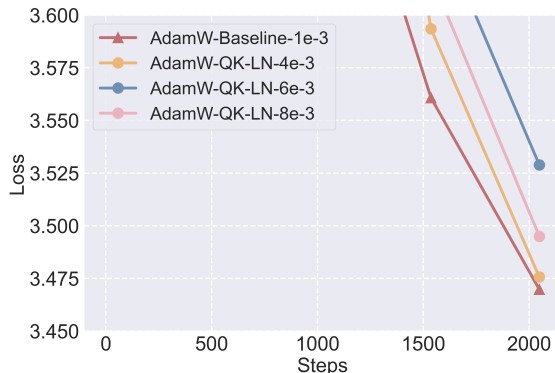

*Figure 8.* QK-Norm leads to performance degradation although improving the feasible learning rates of GPT-2 models pre-training without divergence.

## 5.4. Ablation Study

**Ablation 1: Element-wise Clipping.** Figure 9 reveals that MERIT, even without element-wise clipping, still outperforms AdamW and LAMB in terms of convergence, albeit with a less pronounced improvement. This finding suggests that when we apply the element-wise trust ratio without clipping, certain elements undergo unexpectedly large update steps, which can adversely affect the language model performance. These results underscore the importance of element-wise update clipping in large-batch training scenarios, where update magnitudes tend to be larger compared to standard training conditions. For visualization of element-wise clipping ratios during GPT-2 small training, please refer to Appendix H.

**Ablation 2: Weight-wise Ratio Bound.** The implementation of a weight-wise trust ratio as a lower bound for element-wise ratios aims to mitigate excessively small updates during large-batch training of language models. As illustrated in Figure 9, the application of this lower bound significantly enhances generalization performance, highlighting the importance of balanced updates across different elements. The bounding ratio is given in Appendix K.

**Ablation 3: Element-wise Ratio.** Element-wise trust ratio calculations enhance the generalization capability of language models by providing more robust ratio estimates focusing on local weight structures for individual weight elements. Figure 9 demonstrates the advancement of using an element-wise ratio.

*Figure 9.* Three ablations present obvious performance degradations, which validates the necessity of all three design choices.

**Ablation 4: Norm Choice.** To investigate the impact of the norm choice in the trust ratio computation, we conducted an ablation study where we replaced the weight-wise $l_2$-norm with the max-norm, resulting in the maxLAMB optimizer. This variant excludes the element-wise ratio and clipping components used in MERIT. Results in Table 3 on GPT-2 small and medium show that maxLAMB marginally improves over LAMB but underperforms MERIT, confirming

that MERIT's gains stem from its combined use of element-wise trust ratio and clipping rather than max-norm alone. Appendix K further supports this by demonstrating the critical role of element-wise operations in early training.

*Table 3.* Validation loss comparison across optimizers (AdamW, LAMB, maxLAMB, and MERIT) on GPT-2 small and medium. MERIT achieves the lowest validation loss, demonstrating the effectiveness of its element-wise trust ratio and clipping mechanism.

| Optimizer | GPT-2 Small | GPT-2 Medium |
|---|---|---|
| AdamW | 3.470 | 3.172 |
| LAMB | 3.355 | 3.068 |
| maxLAMB | 3.304 | 3.040 |
| MERIT | 3.280 | 2.982 |

## 6. Conclusion

Accelerating the pre-training of language models heavily relies on large batch techniques. In this study, we present the MERIT optimizer, which integrates max norm and local weight information to compute trust ratios. When applied to the large-batch training of GPT models, MERIT enables larger batch size usage than LAMB and AdamW, while maintaining comparable generalization performance.

## Impact Statement

This paper presents work whose goal is to advance the field of Machine Learning. There are many potential societal consequences of our work, none which we feel must be specifically highlighted here.

## Acknowledgements

Yang You's research group is being sponsored by NUS startup grant (Presidential Young Professorship), Singapore MOE Tier-1 grant, ByteDance grant, ARCTIC grant, SMI grant and Alibaba grant.

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

## A. Models and Hyperparamters Configuration

Table 4. Model Configurations and Peak Learning Rate Under Chinchilla Scaling Law.

| Model | Size | d_model | n_head | depth | Lion lr | Sophia-G lr | AdamW lr | LAMB lr | MERIT lr |
|-------|------|---------|--------|-------|---------|-------------|----------|---------|----------|
| Small | 125M | 768 | 12 | 12 | 1e-4 | 1e-4 | 1e-3 | 1e-2 | 9e-3 |
| Medium | 355M | 1024 | 16 | 24 | 8e-5 | 1e-4 | 4e-3 | 1e-2 | 9e-3 |
| Large | 770M | 1280 | 20 | 36 | 8e-5 | 2e-4 | 2e-3 | 8e-3 | 6e-3 |

In our study, we examine three GPT-2 variants: small, medium, and large, as described by (Radford et al., 2019). The specific configurations for these models are outlined in Table A. We utilize the nanoGPT framework [2] as our codebase. Consistent with nanoGPT's approach, we implement GELU activation functions and omit bias and Dropout (Srivastava et al., 2014) during the pre-training phase.

The GPT-2 models undergo training using the OpenWebText corpus (Gokaslan & Cohen, 2019). We process the text using the GPT-2 tokenizer (Radford et al., 2019). For data organization, we adopt the train-validation split provided by nanoGPT. The training dataset comprises 9 billion tokens, while the validation set contains 4.4 million tokens.

Our training setup employs distributed data-parallel processing with gradient accumulation, allowing for batch sizes of 1K, 4K, and 8K. All model variants are trained using bfloat16 precision. The 125M and 355M parameter models are trained on systems equipped with two H100 GPUs, whereas the 770M parameter models require machines with eight H100 GPUs.

## B. Limitations

**Comprehensive downstream task assessment.** We evaluate large-batch pre-trained models on 7 downstream tasks, which provides valuable but limited insights. A truly comprehensive assessment of language models remains an open research challenge. Our evaluation is further constrained by the modest size of the models studied, which lack advanced capabilities like in-context learning and complex reasoning capability. These limitations indicate the need for caution when extrapolating our findings to larger, more capable models.

**Cross-domain applicability and generalization.** Our study focuses on large language model optimization. However, a truly versatile optimizer should perform well across various domains such as computer vision, reinforcement learning, and multimodal tasks. Due to computational constraints, we have not evaluated the large-batch training performance of our optimizer in these areas. Future work should investigate its efficacy across diverse machine learning paradigms to fully assess its generalizability and potential impact.

**Scaling up to larger language models and datasets.** MERIT has shown promising scalability up to 770M parameter models trained on OpenWebText. While there are no fundamental barriers to scaling further, our comparison with AdamW and LAMB on more extensive models and datasets is constrained by resource limitations. Based on observed improvements in scaling laws and enhanced pre-training stability, we anticipate MERIT to outperform AdamW and LAMB in large-batch training scenarios with larger language models. However, empirical validation of this hypothesis awaits future work with access to greater computational resources.

## C. Relation between learning rate and attention logits

$$W_Q^{(t+1)} = W_Q^{(t)} - \eta \nabla_{W_Q} L, \quad W_K^{(t+1)} = W_K^{(t)} - \eta \nabla_{W_K} L$$

$$Q_i^{(t+1)} = X_i W_Q^{(t+1)} = X_i(W_Q^{(t)} - \eta \nabla_{W_Q} L) = X_i W_Q^{(t)} - \eta X_i \nabla_{W_Q} L,$$

$$K_j^{(t+1)} = X_j W_K^{(t+1)} = X_j(W_K^{(t)} - \eta \nabla_{W_K} L) = X_j W_K^{(t)} - \eta X_j \nabla_{W_K} L$$

---

[2]https://github.com/karpathy/nanoGPT

$$\text{Logits}_{ij}^{(t+1)} = \frac{Q_i^{(t+1)} \cdot (K_j^{(t+1)})^\top}{\sqrt{d_k}}$$

$$= \frac{Q_i^{(t)} \cdot (K_j^{(t)})^\top}{\sqrt{d_k}} - \eta \frac{Q_i^{(t)} \cdot (X_j \nabla_{W_K} L)^\top}{\sqrt{d_k}} - \eta \frac{(X_i \nabla_{W_Q} L) \cdot K_j^{(t)\top}}{\sqrt{d_k}} + \eta^2 \frac{(X_i \nabla_{W_Q} L) \cdot (X_j \nabla_{W_K} L)^\top}{\sqrt{d_k}}$$

Assuming that the gradients $\nabla_{W_Q} L$ and $\nabla_{W_K} L$ are not dependent on $\eta$ (which is typical in gradient descent), the change in the max attention logit is linearly proportional to $\eta$ because $\eta^2$ terms are negligible compared to $\eta$ terms:

$$\text{Max Logit}^{(t+1)} \approx \text{Max Logit}^{(t)} - \eta \cdot C + \mathcal{O}(\eta^2) \propto \eta$$

## D. Influence of Max Norm On Max Attention Logit

Given:
$$||W_Q||_m = M_Q \text{ and } ||W_K||_m = M_K$$

Each element in $W_Q$ and $W_K$ satisfies:

$$0 \leq |W_{Q,i,j}| \leq M_Q \text{ and } 0 \leq |W_{K,i,j}| \leq M_K$$

Each element in Q and K can be bounded as:

$$Q_{i,k} = \sum_{m=1}^{n} X_{i,m} W_{Q,m,k} \leq \sum_{m=1}^{n} |X_{i,m}| M_Q = M_Q \sum_{m=1}^{n} |X_{i,m}| = M_Q \cdot C_X$$

Similarly,
$$K_{j,k} \leq M_K \cdot C_X$$

where $C_X = \sum_{m=1}^{n} |X_{i,m}|$ is a constant representing the sum of absolute values in the input embeddings for a token and $n$ is the hidden size. The attention logit is:

$$Logit_{i,j} = \sum_{k=1}^{d} Q_{i,k} K_{j,k} / \sqrt{d} \leq \sum_{k=1}^{d} (M_Q \cdot C_X)(M_K \cdot C_X) / \sqrt{d} = \sqrt{d} \cdot M_Q M_K C_X^2$$

Therefore,
$$Logit_{i,j} \leq \sqrt{d} \cdot M_Q M_K C_X^2.$$

Moreover, In decoder-only model implementations, each layer includes LayerNorm(Ba et al., 2016), which normalizes $X$ to follow a Gaussian distribution. This normalization effectively establishes upper bounds $C_X$ on $X$'s values.

Thus,

$$Logit_{i,j} \leq \sqrt{d} \cdot M_Q M_K C_X^2 = \mathcal{O}(dn^2 \cdot M_Q M_K).$$

This proof demonstrates that controlling the max norm of $W_Q$ and $W_K$ effectively constrains the upper bound of the max attention logit, which is crucial for large-batch training stability.

# E. Huge Difference between $l_2$ Norm and Max Norm

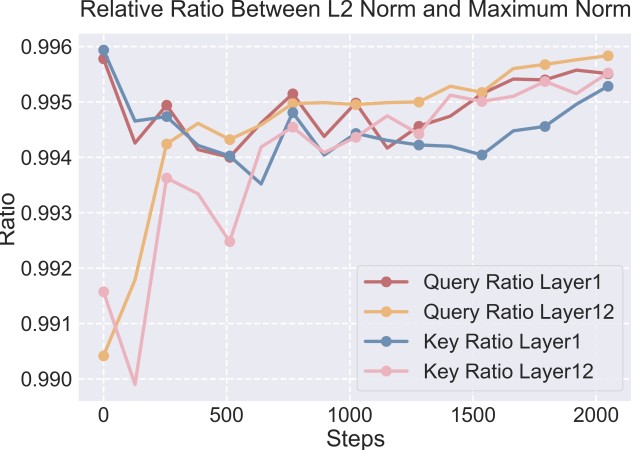

*Figure 10.* Relative ratios between max and $l_2$ norm of $W_K/W_Q \in \mathbb{R}^{1024 \times 1024}$ in GPT-2 medium. Ratio is calculated as $(\|W\| - \|W\|_m)/\|W\|$, in which $\| \cdot \|$ and $\| \cdot \|_m$ denote $l_2$-norm and max norm.

Figure 10 illustrates the relative ratio between $l_2$ norm and maximum norm for query and key weights across different layers (Layer 1 and Layer 12) during 2K training steps. The ratios consistently remain high, hovering around 0.99-0.996, which reveals a critical insight: the gap between $l_2$ norm and max norm is huge. When optimizing using only $l_2$ norm, the weight updates affect all parameters globally but fail to constrain the max value of query/key weight matrix.

# F. Max attention logit in small-batch training

Figure 11 shows the distribution of max attention logits in small-batch (512) training of the GPT-2 medium model using the same chinchilla scaling law setting. Notably, these max attention logits are significantly lower than those observed in large-batch training scenarios. This reduction suggests that the attention outputs are more evenly distributed, which typically leads to improved training convergence and generalization performance.

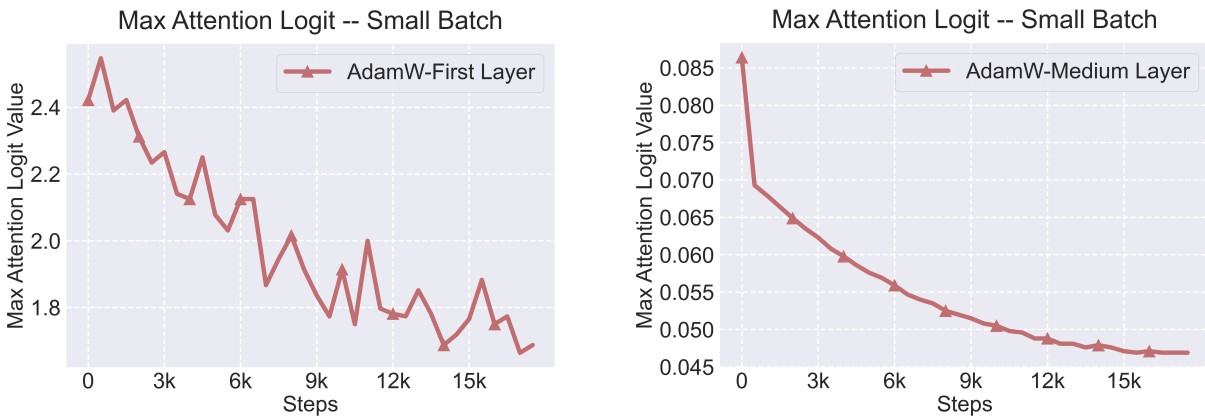

*Figure 11.* Max attention logit of self-attention layers during the small-batch training of GPT-2 medium model using three optimizers. (a) Max Attention Logit of first self-attention layer. (b) Max Attention Logit of medium self-attention layer.

# G. Zero-shot evaluation on LAMBADA and WikiText

**Zero-shot Evaluation.** The improved validation loss leads to better downstream task performance, as demonstrated in Figure 12. When comparing models with equal pre-training steps, the GPT-2 variants trained using MERIT consistently

outperform those using LAMB and AdamW in zero-shot accuracy across most subtasks.

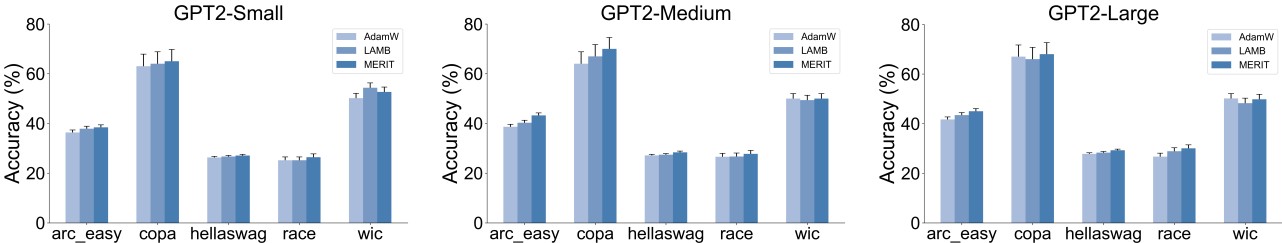

*Figure 12.* Zero-shot evaluation on SuperGLUE benchmark. Given an equivalent number of training steps, models that undergo large-batch pre-training using MERIT exhibit higher accuracy than those pre-trained with AdamW and LAMB on most tasks.

## H. Clipping Ratio in GPT-2 Small Large-batch Training

Figure 13 presents the element-wise clipping ratio in GPT-2 small training setting (2B tokens) for the 1st, 6th, and 12th layers.

The analysis of clipping effects across different layers reveals distinct patterns in gradient update behavior during training. The input layer (Layer 1) maintains near-zero clipping ratios throughout, suggesting that early-layer gradient updates rarely require adjustment. In contrast, the middle layer (Layer 6) experiences more substantial clipping, with ratios peaking at 12% during later training stages. The output layer (Layer 12) shows minimal clipping, with ratios reaching only 0.25% at maximum.

This layered pattern demonstrates that the clipping mechanism primarily influences the middle layers, leaving input and output layers unaffected. This suggests that the clipping mechanism primarily stabilizes middle layers rather than applying a uniform constraint across all layers. Most gradient updates maintain their original direction, with the most significant stabilization occurring in middle layers where feature representations undergo refinement.

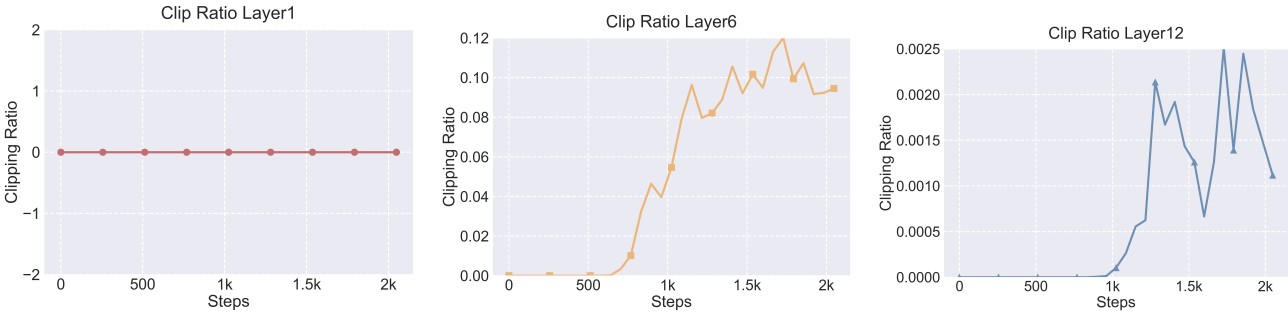

*Figure 13.* Element-wise clipping ratio for different layers of GPT-2 small during large-batch training.

## I. Visualization of Max-Attention Logits

Figure 14 presents the max attention logits (MAL) across all layers in the training of GPT-2 medium, observing that MERIT consistently stabilizes training by mitigating extreme MAL spikes. In shallow layers (1–8), both MERIT and LAMB effectively reduce the high MAL values seen with AdamW. In mid-depth layers (9–17), MERIT's max-norm control further mitigates the MAL spikes present in LAMB. In deeper layers (18–24), MERIT exhibits comparable or marginally higher MAL values, though these remain significantly lower than in mid-layers and have negligible effects on convergence. These findings highlight MERIT's effectiveness in controlling destabilizing attention logits, especially in crucial mid-layers, facilitating stable large-batch training and better convergence.

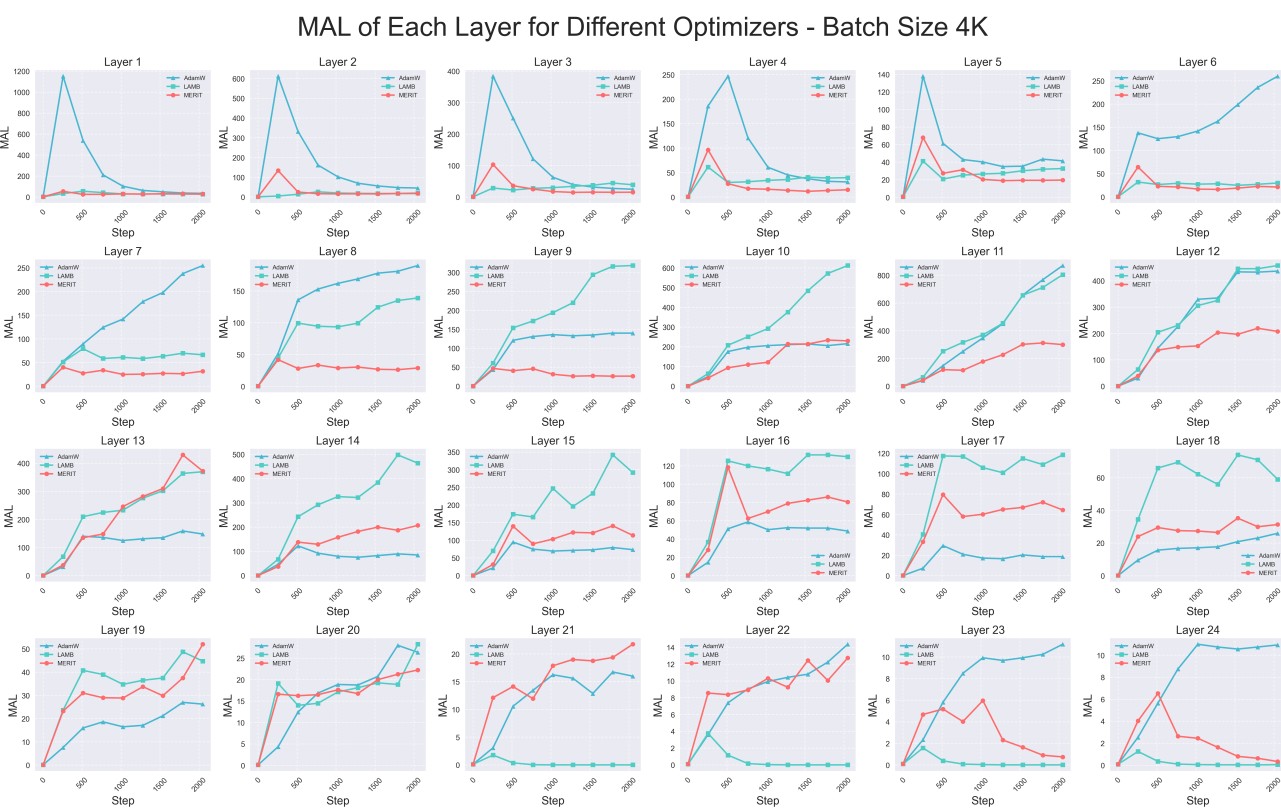

*Figure 14.* Max attention logit of self-attention layers during the large-batch training of GPT-2 medium using three optimizers of all layers

## J. Convergence Proof of Theorem 1

**Proof.** We study how MERIT-W converges across different minibatch sizes. To begin, let's review the update equation of MERIT-W

$$w_{t+1}^{(i)} = w_t^{(i)} - \eta_t \cdot \|w_t^{(i)}\|_m \frac{u_t^{(i)}}{\|u_t^{(i)}\|_m} \tag{5}$$

for all $i \in [h]$.

Since the function $f$ is $L$-smooth, we obtain the following:

$$f(w_{t+1}) \leq f(w_t) + \langle \nabla_i f(w_t), w_{t+1}^{(i)} - w_t^{(i)} \rangle + \sum_{i=1}^h \frac{L_i}{2} \|w_{t+1}^{(i)} - w_t^{(i)}\|^2$$

$$\leq f(w_t) \underbrace{-\eta_t \sum_{i=1}^h \sum_{j=1}^{d_i} (\|w_t^{(i)}\|_m) \times \left( [\nabla_i f(w_t)]_j \times \frac{u_t^{(i,j)}}{\|u_t^{(i)}\|_m} \right)}_{T_1} + \sum_{i=1}^h \frac{L_i d_i \alpha_u^2 \eta_t^2}{2} \tag{6}$$

The above inequality follows from the Lipschitz continuity of the gradient. We bound term $T_1$ in the following manner:

$$T_1 \leq -\eta_t \sum_{i=1}^h \sum_{j=1}^{d_i} \|w_t^{(i)}\|_m \times \left( [\nabla_i f(w_t)]_j \times \frac{u_t^{(i,j)}}{\|u_t^{(i)}\|_m} \right)$$

$$\leq -\eta_t \sum_{i=1}^h \sum_{j=1}^{d_i} \sqrt{\frac{1-\beta_2}{G^2 2\log(d_i)}} \left( \|w_t^{(i)}\|_m \times [\nabla_i f(w_t)]_j \times g_{t,j}^{(i)} \right)$$

$$-\eta_t \sum_{i=1}^h \sum_{j=1}^{d_i} \left( \|w_t^{(i)}\|_m \times [\nabla_i f(w_t)]_j \times \frac{u_t^{(i,j)}}{\|u_t^{(i)}\|_m} \right) \mathbf{1}(\text{sign}([\nabla_i f(w_t)]_j) \neq \text{sign}(u_t^{(i,j)}))$$

This follows from the fact that $\|u_t^{(i)}\|_m \leq \sqrt{\frac{2\log(d_i)}{1-\beta_2}}$ and $\sqrt{v_t} \leq G$. If $\beta_2 = 0$, then $T_1$ can be bounded as follows:

$$T_1 \leq -\eta_t \sum_{i=1}^h \sum_{j=1}^{d_i} \sqrt{\frac{1}{2\log(d_i)}} \left( \|w_t^{(i)}\|_m \times [\nabla_i f(w_t)]_j \right)$$

$$-\eta_t \sum_{i=1}^h \sum_{j=1}^{d_i} \left( \|w_t^{(i)}\|_m \times [\nabla_i f(w_t)]_j \times \frac{u_{t,j}^{(i)}}{\|u_t^{(i)}\|_m} \mathbf{1}(\text{sign}([\nabla_i f(w_t)]_j) \neq \text{sign}(u_{t,j}^{(i)})) \right)$$

The rest of the proof for $\beta_2 = 0$ is similar to argument for the case $\beta_2 > 0$, which is shown below.
Taking expectation, we have the following:

$$\mathbb{E}[T_1] \leq -\eta_t \sum_{i=1}^h \sum_{j=1}^{d_i} \sqrt{\frac{1-\beta_2}{G^2 2\log(d_i)}} \mathbb{E}\left[ \|w_t^{(i)}\|_m \times \left( [\nabla_i f(w_t)]_j \times g_{t,j}^{(i)} \right) \right]$$

$$-\eta_t \sum_{i=1}^h \sum_{j=1}^{d_i} \mathbb{E}\left[ \|w_t^{(i)}\|_m \times \left( [\nabla_i f(w_t)]_j \times \frac{u_{t,j}^{(i)}}{\|u_t^{(i)}\|} \right) \mathbf{1}(\text{sign}([\nabla_i f(w_t)]_j) \neq \text{sign}(g_{t,j}^{(i)})) \right]$$

$$\leq -\eta_t \sum_{i=1}^h \sum_{j=1}^{d_i} \sqrt{\frac{1-\beta_2}{G^2 2\log(d_i)}} \mathbb{E}\left[ \left( \|w_t^{(i)}\|_m \times [\nabla_i f(w_t)]_j \times g_{t,j}^{(i)} \right) \right]$$

$$+\eta_t \sum_{i=1}^{h} \sum_{j=1}^{d_i} \mathbb{E}\left[\alpha_u |[\nabla_i f(w_t)]_j| \mathbf{1}(\text{sign}([\nabla_i f(w_t)]_j) \neq \text{sign}(g_{t,j}^{(i)}))\right]$$

Using the bound on the probability that the signs differ, we get:

$$\mathbb{E}[T_1] \leq -\eta_t \alpha_l \sqrt{\frac{h(1-\beta_2)}{G^2 2\log(d)}} \|\nabla f(w_t)\|^2 + \eta_t \alpha_u \sum_{i=1}^{h} \sum_{j=1}^{d_i} \frac{\sigma_{i,j}}{\sqrt{b}}.$$

Substituting the above bound on $T_1$ in equation 6, we have the following bound:

$$\mathbb{E}[f(w_{t+1})] \leq f(w_t) - \eta_t \alpha_l \sqrt{\frac{h(1-\beta_2)}{2G^2 \log(d)}} \|\nabla f(w_t)\|^2 + \eta_t \alpha_u \frac{\|\tilde{\sigma}\|_1}{\sqrt{b}} + \frac{\eta_t^2 \alpha_u^2 d \|L\|_1}{2}$$

Summing the above inequality for $t = 1$ to $T$ and using telescoping sum, we have the following inequality:

$$\mathbb{E}[f(w_{T+1})] \leq f(w_1) - \eta_t \alpha_l \sqrt{\frac{h(1-\beta_2)}{2G^2 \log(d)}} \sum_{t=1}^{T} \mathbb{E}[\|\nabla f(w_t)\|^2] + \eta T \alpha_u \frac{\|\tilde{\sigma}\|_1}{\sqrt{b}} + \frac{\eta^2 \alpha_u^2 dT}{2} \|L\|_1.$$

Rearranging the terms of the above inequality, and dividing by $\eta T \alpha_l$ we have:

$$\sqrt{\frac{h(1-\beta_2)}{2G^2 \log(d)}} \frac{1}{T} \sum_{t=1}^{T} \mathbb{E}[\|\nabla f(w_t)\|^2] \leq \frac{f(x_1) - \mathbb{E}[f(w_{T+1})]}{T \eta \alpha_l} + \frac{\alpha_u \|\tilde{\sigma}\|_1}{\alpha_l \sqrt{b}} + \frac{\eta d \alpha_u^2}{2\alpha_l} \|L\|_1$$

$$\leq \frac{f(w_1) - f(w^*)}{T \eta \alpha_l} + \frac{\alpha_u \|\tilde{\sigma}\|_1}{\alpha_l \sqrt{b}} + \frac{\eta d \alpha_u^2}{2\alpha_l} \|L\|_1$$

## K. Weight-wise Bounding Ratio Distribution

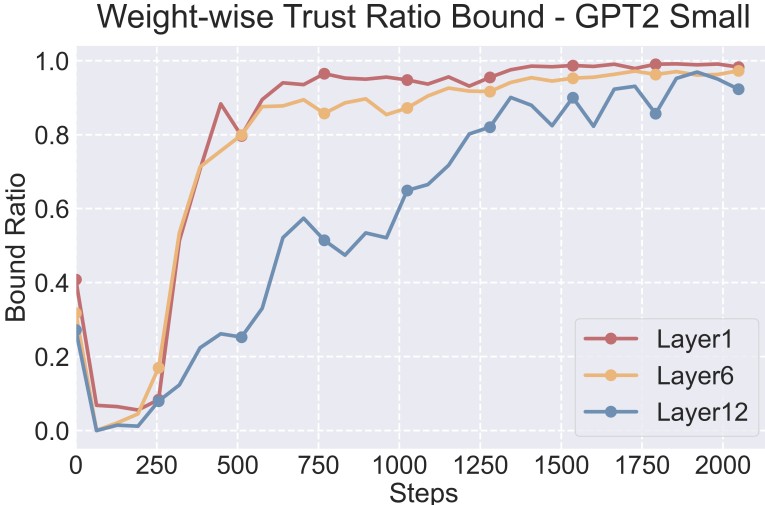

*Figure 15.* The trigger ratio of the weight-wise trust ratio lower bound during the large-batch training of GPT-2 Medium.

Figure 15 further demonstrates that this lower bound becomes particularly crucial in the latter stages of training. This observation indicates that maintaining a minimum update magnitude grows increasingly important as the model nears convergence.

