# OpenReview forum: "MERIT: Maximum-normalized Element-wise Ratio for Language Model Large-batch Training"
_ICML.cc/2025/Conference — ICML 2025 poster_

### Official Review · Reviewer_ybUy · 2025-02-28

**Overall Recommendation:** 4

**Summary:**

This paper addresses the issue of designing an effective optimizer for large-batch training of LLMs.  Normally, as batch size, B, increases, we also increase the LR (often linearly with B, sometimes with the square root of B).  But at some point, optimizers struggle with such high LRs, and become unstable or fail to train well.  Techniques to address this have been proposed previously, in particular LAMB from You et al. (2020).  Meanwhile, people training LLMs have noticed that very unbalanced QK attention logits have contributed to instability in Transformers, and also hyperparameter sensitivity.  Putting these two issues together, this paper proposes that perhaps it’s this “attention entropy collapse” issue that is preventing large-batch training from working well.  They propose MERIT to prevent this.  It’s basically a version of LAMB that scales and normalizes by the maximum weight (max-norm), rather than the L2 norm of the layerwise weights as in LAMB.  MERIT does seem to reduce the max attention logit in the first layer (where it seems to grow particularly high), but so does LAMB, actually.  But MERIT does seem to reduce it more in Layer 12 in a GPT2 medium model.  In experiments at different GPT2 model scales, trained to around 20 TPP, where every optimizer is tuned to their optimum max LR with grid search, MERIT shows small advantages over LAMB, and LAMB over AdamW.

**Claims And Evidence:**

[013, left]: Claim: “Large-batch training ... poses challenges in generalization”
- I think there is a misunderstanding about what is meant by the “generalization gap.”  Larger batches eventually perform worse (on train and test error) for the same total number of training examples, because the gradient approximation can only improve so much as B increases – even full-batch GD wouldn’t work much better than just using a very large batch size (but would consume a lot more total examples).  This phenomenon is illustrated in your Figure 1.  Note this has nothing to do with “unseen data” – you would see this same result if all settings instead had a common target *training* loss.  The idea that larger batches can hurt generalization (performance on unseen data) was an idea that was prevalent during the 2010s decade, and I believe has been well and truly refuted, e.g., see Shallue et al., McCandlish et al.  E.g., Shallue et al report, “We find no evidence that larger batch sizes degrade out-of-sample performance.”
- unless you have a specific reason to think your models are failing to generalize, I would probably re-write your paper not to mention “generalization” at all.  E.g., you can just say “performance” rather than “generalization performance” or “validation loss” as needed.
- I also find that this paper does not seem to understand that fundamentally, as you increase batch sizes, you get redundant information, so very large batches are just inherently going to not work as well as smaller ones given the same total data (although they will do better with the same number of STEPS/batches).  Sentences like “As a result [of attention entropy collapse], AdamW-based large-batching leads to a worse generalization performance than small batches” are attributing all the degradation to training dynamics, but fundamentally, large batches will just not work as well.

[016, left]: Claim: the reason prior optimizers fail to work well with large batches is due to “sharp increase of max attention logit”.
- after reading this paper, I am not convinced of this.  Perhaps MERIT is improving on LAMB and AdamW for some other reason?
- Shouldn’t there be a plot in the main paper showing how the max attention logit changes as we increase the batch size?  If you could show that this was specifically a problem with large-batches, and that MERIT does better with larger batches, and MERIT is kind of designed to address this problem, then I think this claim would be more valid.
- Note: I would REALLY like the paper to show this.

[029, left]: “MERIT ... leverages the max-norm to calculate the trust ratio to constrain the max attention logit more effectively.”
- I also found the evidence for this lacking.  We see results in Figure 2 for two specific layers, of one model.  Can’t we get these statistics over all the layers of all the models?
- If you could show that MERIT reduces the max attention logits more over ALL the layers, that would go a long way to showing that this really was the problem.
- Note: I would REALLY REALLY like to see this too.

**Essential References Not Discussed:**

[027, right]: “Training large language models with large batches typically encounters one main issue: research has shown that training with large batches often leads to models performing poorly on unseen data” [citation needed]

[087, right]: ‘dubbed the “generalization gap”’ [citation needed]

**Experimental Designs Or Analyses:**

Figure 1 is ultimately quite surprising to me: AdamW, MERIT, and LAMB take EXACTLY the same number of steps to reach the target validation loss with the same batch size, across three batch sizes?  I mean... wouldn’t it depend on their hyperparameters to some extent?  It’s just very surprising.  And also, it is suprising that MERIT shows absolutely no degradation at the largest batch size, but AdamW and LAMB do (while AdamW and LAMB show no degradation at all for the smaller batch sizes).  This figure is so perfect that it’s suspicious, is what I’m saying.  Even just cutting off the plot before larger batch sizes... it’s okay if perfect scaling doesn’t continue!  There is no optimizer that can scale perfectly forever!
- also, note my above point: we need to know the number of steps in advance in order to set the decay schedule... so how can you even determine how many steps it takes to reach a target validation loss?

**Methods And Evaluation Criteria:**

For Figure 1, since you use a LR decay schedule to 10% of the peak LR, how do you train to exactly the same validation loss?  This relates to m concern: Figure 1 is suspiciously perfect.

[188, left] “the proposed method gives larger updates to extreme values of weights”
- I suppose LAMB would do this too.  I don’t really get why this is a good idea, or why it prevents anything.
- Like, I understand why you might want to normalize the update by the max-norm (denominator of ratio), but why you want the updates to be on the scale of the max-norm is less obvious to me.
- It seems to me you want to prevent having extreme max-norms to begin with, not behave a certain way once they are there?
- But anyways, it does seem like this helps a little bit (Figure 2b), but it’s not clear at this point that this is really significant for results.

For 5.1 and 5.2, is the idea that we are already picking a very large batch size, to the point where we expect MERIT > AdamW (i.e., unlike the batch sizes in Figure 1?).  Might be good to say this.  Like, based on Figure 1, it seems you picked the exact smallest batch size such that AdamW, LAMB, and MERIT behave differently at all.

Curvature of convergence point:
- I’m pretty sure we’re not training GPT2 models to a convergence point.  Do you mean the point where you stopped training?
- I don’t believe that either AdamW or MERIT has any trouble generalizing from training to test data.  You could check this quite easily, right?  E.g., scatter plot validation loss vs. train loss for all your training runs with each optimizer.

Regarding the Figure 6(b) results:
- Which GPT2 model is this?
- Why not try QK-LN-1e-3????  And if that’s better, why not QK-LN-8e-4?  I’m not convinced the LR is tuned here.
- How can we conclude it’s a problem with large batches?  Did we try this for small batches and QK-LN does work fine?

**Other Comments Or Suggestions:**

Intro:

Overall this was good.

Might have been helpful to point out in the intro that “high max entropy logit” is also known as “attention entropy collapse” and techniques like QK-layernorm have been proposed to address it, but, I assume, not in the context of large batches?

In the intro, I could have used some help understand what MERIT will *do* exactly.  It uses max norms, it calculates trust ratios, but how does it actually play a role as an optimizer?  Does it precondition, or scale, or normalize weight updates?  Does it reguarlize training in some way?  Assume the reader understands how optimizers work, but may not be familiar with LAMB, for example.

Figure 1: caption: “further increasing the batch sizes offers no additional advantage”
- just to clarify, this plot does NOT show that phase, right?  Even for vanilla AdamW, increasing the batch size does decrease the number of steps, it looks like this holds even to 2^13.  Having this statement in the caption might lead one to think the figure shows this no-additional-advantage phase.
- just to emphasize this point: in some cases, if we want a good LLM ASAP, it might be worth using more total training examples, provided we can get our results faster (fewer steps).  So there's a time "advantage" even beyond perfect scaling.

[129, right]: that’s a run-on sentence, not grammatically correct, you should probably divide into multiple sentences, and I think you mean “expressive” ability.

Preliminary → Preliminaries ?

Equation (1):
- is that λ the weight decay?  This does not seem to be defined anywhere in the paper.
- I’m confused, you’re saying this whole term is the trust ratio?  Or is the ratio not just the ratio of norms, and then we apply this ratio to the weight update???  And this effectively scales the LR?  Because otherwise, as written, this value R looks like a vector to me.
- Could we preface this whole section by saying, “since large-batch training uses large LRs, LAMB proposes to re-scale these LRs at each layer to mitigate the effects of very large or very small gradients/updates, and to ensure that the updates that are made are on the scale of the original weights” or something like that.  The connection to large-batch training is just stated, but not *explained* in the current version.

Figure 4 caption says “final validation loss” but the figure has intermediate validation losses as well.  I understand you report the final losses in the caption.  Maybe you need to say, “Final validation losses are:” and then continue from there.

Perhaps Figure 6 should just be two separate figures?

**Other Strengths And Weaknesses:**

The paper was very well written, and the experiments were conducted in a very convincing way, and training to 20 tokens-per-parameter following Chinchilla was an excellent decision, and grid-searching the LR was very important and well done.

**Questions For Authors:**

See other comments

**Relation To Broader Scientific Literature:**

This paper builds fairly directly on LAMB, but incorporating recent findings regarding attention entropy collapse.

I think the comparison with QK-layernorm is very important, as it’s an independent mechanism designed to address this problem.  But unfortuntely, I do not think the comparison is very convincing.

[035, right]: This paper “identifies” the problem of large batches: Sharp increase in the max attention logit in attention layers using AdamW
- I am a little confused.  Since the problem is only now identified, are you saying that LAMB partially addresses this problem *without identifying it as a problem*.  At this point in the paper, it might be better to say that LAMB was designed to do X, but, although it was not the specific intention, it does help with attention entropy collapse, although less so than MERIT.

**Theoretical Claims:**

4.4 Convergence analysis:
- You need to explain what you are intending to show, and then explain exactly what you found: is the convergence rate of MERIT better than AdamW or LAMB?  Under what specific assumptions?  With the current discussion, I have trouble understanding why this analysis is useful.  You don’t mention this analysis as a contribution in your introduction, or refer to it in your conclusion.  So what’s the point?  Is it something that should be in the appendix?

---

> ### Author Rebuttal · Authors · 2025-03-31
>
> We sincerely thank the reviewer ybUy for the valuable questions and comments. All suggestions on claims and layout revisions will be revised in the updated paper. For the concerns and questions, here are our responses:
>
> **Q1: Need for the figure plotting how the max attention logit changes as we increase the batch size.**
>
> **A1:** Thank you for the suggestion. We ran extensive experiments to find the best setting for each optimizer and batch size, and then plotted the max attention logit (MAL) changes accordingly.
>
> - **Peak MAL** means the maximum MAL across all layers and training steps.
> - **Layers with MAL > 100** counts how many layers exceed a MAL of 100.
>
> Pls check full plots at: https://anonymous.4open.science/r/MAL-F3FE.
>
> Peak MAL (Val Loss in parentheses):
>
> |Optimizer|512|1K|2K|4K|
> |:-:|:-:|:-:|:-:|:-:|
> |AdamW| 326 (2.830)|664 (2.871)|604 (2.976)|1152 (3.172)|
> |LAMB|464 (2.872)|352 (2.864)|480 (2.919)|804 (3.068)|
> |MERIT|172 (2.841)|286 (2.853)|262 (2.890)|458 (2.982)|
>
> **Analysis:**
>
> - Peak MAL generally correlates with higher validation loss.
> - All optimizers show an increasing trend of Peak MAL as batch size grows.
> - MERIT exhibits the slowest growth in Peak MAL, indicating better stability.
>
> Layers with MAL > 100:
>
> |Optimizer|512|1K|2K|4K|
> |:-:|:-:|:-:|:-:|:-:|
> |AdamW|8|9|12|14|
> |LAMB|3|8|10|10|
> |MERIT|3|4|6|9|
>
> **Analysis:** The increasing batch size leads to more layers that have high MAL, which affects the training stability.
>
> **Conlusion:** MERIT reduces both Peak MAL and the number of layers MAL > 100 as batch size increases, enabling more stable and effective large-batch training.
>
> **Q2: Need for the figure plotting how the max attention logit changes across all layers.**
>
> **A2:** Thanks for your comment. We plotted the full MAL distribution across all layers for GPT-2 Medium with batch size 4K. See details at: https://anonymous.4open.science/r/MAL-4K-2K-277A/.
>
> **Analysis:**
>
> - In layers 1-8, where AdamW shows high MAL, MERIT and LAMB effectively suppress it.
> - In layers 9-17, MERIT reduces LAMB's MAL spikes via max-norm control.
> - In deeper layers (18-24), MERIT shows similar or slightly higher MAL, but the values are much smaller than in layers 1-17 and have minimal impact on convergence.
>
> Crucially, MERIT shows significantly lower peak MAL across all layers than LAMB:
>
> |Optimizer|Peak MAL|Val loss|
> |:-:|:-:|:-:|
> |AdamW|1152 (layer1)|3.172|
> |LAMB|804 (layer11)|3.068|
> |MERIT|458 (layer13)|2.982|
>
> **Conclusion**: MERIT more effectively reduces excessive MAL peaks in critical layers (9-17), leading to better model stability and improved validation loss under large-batch training.
>
> **Q3: Figure 1 is suspiciously perfect.**
>
> **A3:** Thank you for your question. In Figure 1, we approximated that an optimizer "achieves" the target loss if its final validation loss falls within $\pm$0.05 of the expected range (e.g., 3.23-3.33 in the left plot). This simplification was made to reflect training efficiency trends under limited computing resources without enough grid search, which is why the figure may appear perfect -  apologies for any confusion.
>
> Due to time constraints, we could not determine the exact step counts for all configurations. To provide a more rigorous comparison, we conducted extensive experiments and report the best validation loss achieved at each batch size:
>
> **GPT-2 Small:**
>
> | Optimizer | bsz = 128 | bsz = 256 | bsz = 512 | bsz = 1K |
> |:-:|:-:|:-:|:-:|:-:|
> |AdamW|3.172|3.194|3.288|3.470|
> |LAMB|3.189|3.176|3.247|3.355|
> |MERIT|3.205|3.182|3.214|3.280|
>
> **GPT-2 Medium:**
>
> |Optimizer|bsz=512|bsz=1K|bsz=2K|bsz=4K|
> |:-:|:-:|:-:|:-:|:-:|
> |AdamW|2.830|2.871|2.970|3.172|
> |LAMB|2.870|2.864| 2.919| 3.068|
> |MERIT|2.841|2.853|2.890|2.982|
>
> **Conclusion:** AdamW shows clear performance degradation as batch size increases. In contrast, MERIT maintains strong performance and its gains over LAMB grow with batch size-highlighting MERIT's scalability and robustness in large-batch training.
>
> **Q4: Figure 6(b) results.**
>
> **A4:** Thank you for your question. We used GPT2-Small for the experiments in Figure 6(b). To further examine the effect of QK-LayerNormalization, we conducted more extensive tests:
>
> **Large-Batch Setting (bsz = 1K):**
>
> |Method|8e-4|1e-3|2e-3|4e-3|6e-3|8e-3|1e-2|
> |:-:|:-:|:-:|:-:|:-:|:-:|:-:|:-:|
> |AdamW + QK Norm|3.608|3.593|3.524|3.475|3.527|3.496|3.671|
>
> Compared to the AdamW baseline (loss = 3.470), QK Norm enables higher learning rates without divergence. However, it consistently results in worse validation loss in large-batch training, indicating that its stability benefits do not translate into better optimization outcomes under large-batch training.
>
> **Small-Batch Setting (bsz = 128):**
>
> |Method|2e-3|4e-3|6e-3|
> |:-:|:-:|:-:|:-:|
> |AdamW + QK Norm|3.171|**3.167**|3.182|
>
> QK Norm slightly improves small-batch performance (baseline loss = 3.172), suggesting its weaker effect in large-batch settings may stem from fewer steps or the change in training dynamics.

---

> > ### Comment · Reviewer_ybUy · 2025-04-02
> >
> > Thanks for your rebuttal, and for addressing my concerns.
> >
> > Looking over all the reviews and rebuttals, I would just say that I think this is a fairly decent scale for conducting experiments, especially since they require hyperparameter tuning.  I personally think LLM showing gains for new techniques on at least 20-tokens-per-parameter (compute-optimal) training is sufficient.  If we actually need larger scales and higher TPP than that, I think we have a fairness and equity problem in empirical ML research.

---

> > > ### Author Response · Authors · 2025-04-03
> > >
> > > We would like to express our sincere gratitude to reviewer ybUy for acknowledging our work and providing constructive suggestions. We will evaluate the performance of our proposed optimizer in larger-scale settings in future work.

---

### Official Review · Reviewer_QXND · 2025-03-07

**Overall Recommendation:** 2

**Summary:**

This paper proposes a new optimization algorithm MERIT for tackling the large-batch training of language models. The method normalizes the update scale based on the maximum norm of both column and row statistics of the update. Empirical study shows that MERIT with large batch size achieves similar performance as Adam with standard batch size, and exhibits slightly faster convergence than baseline algorithms including Adam, Lion, Sophia-G, etc.

**Claims And Evidence:**

* **Why MERIT is beneficial to large-batch training.** The paper argues that MERIT benefits large-batch training by reducing the maximum attention logit, which is achieved by utilizing the maximum norm rather than $l_2$ norm used in LAMB. However, the magnitude of attention logit is determined by the weight norm of query and key matrices. It is unclear why adjusting the update scale will constrain the norm of these matrices. The link between the algorithmic design and the reduction of maximum attention logits is missing.

* **Justification on scaling to large batch size.** Since MERIT targets for large-batch training, it is crucial to plot the validation loss of MERIT across different batch sizes, and show that its convergence and downstream performance indeed scales with the batch size. The comparison between the baseline methods such as LAMB and Adam should be presented as well (e.g. the Table 2 of the [1]).

[1] Large Batch Optimization for Deep Learning: Training BERT in 76 minutes

**Essential References Not Discussed:**

The paper has discussed the most related works.

**Experimental Designs Or Analyses:**

**Effect of maximum norm.** Based on Figure 7, the performance improvement given by clip operation and element-wise ratio are marginal. The main benefit comes from the weight-wise ratio, which is a feature from the LAMB optimizer. Hence, it seems that the main difference between MERIT and LAMB is to substitute of $l_2$  norm into maximum norm. However, the ablation study on the effect of the norm choice is missing.

**Methods And Evaluation Criteria:**

Yes. The algorithm is based on a seminal work on large-batch training named LAMB. The setting of GPT-2 pre-training is standard.

**Other Comments Or Suggestions:**

The paper's presentation can be improved. For instance, it would be more clear to present the ablation results seperately in a table, rather than a figure of grid.

**Other Strengths And Weaknesses:**

Please see my comments above.

**Questions For Authors:**

Please see my comments above.

**Relation To Broader Scientific Literature:**

The paper contributes machine learning field by introducing a large-batch training method.

**Theoretical Claims:**

The proof is based on strong assumptions such as bounded gradient.

---

> ### Author Rebuttal · Authors · 2025-03-31
>
> We sincerely thank the reviewer QXND for the valuable questions and comments. For the concerns and questions, here are our responses:
>
> **Q1: Why MERIT is beneficial to large-batch training?**
>
> **A1:** Thanks for the point. We have provided a theoretical analysis of why the max-norm is efficient in limiting the increasing of max norm of query/key weights:
>
> Given the query weight $W \in m \times n$ and the updating of MERIT:
>
> $$
> W_{t+1} = W_t - \eta_t \cdot \frac{||W_t||_m}{||U_t||_m} \cdot U_t.
> $$
>
> Then,
>
> $$
> ||W_{t+1}||_m = ||W_t - \eta_t \cdot \frac{||W_t||_m}{||U_t||_m} \cdot U_t||_m \leq ||W_t||_m + \eta_t \cdot \frac{||W_t||_m}{||U_t||_m} \cdot ||U_t||_m = ||W_t||\_m + \eta_t \cdot ||W_t||_m.
> $$
>
> As for LAMB, the upper bound is:
>
> $$
> ||W_{t+1}||_m = ||W_t - \eta_t \cdot \frac{||W_t||_2}{||U_t||_2} \cdot U_t||_m \leq ||W_t||_m + \eta_t \cdot \frac{||W_t||_2}{||U_t||_2} \cdot ||U_t||_m \leq ||W_t||\_m + \eta_t \cdot ||W_t||_2.
> $$
>
> It is obvious that  $||W_t||_m \leq ||W_t||_2$ in most cases and usually $||W_t||_m << ||W_t||_2$. Thus, MERIT provides a tighter bound for max norm of query/key weights compared with LAMB.
>
> In Appendix D, we analyzed how the max norms of query and key weights affect max attention logits (MAL). In the visualization(https://anonymous.4open.science/r/MAL-4K-2K-277A/), MERIT effectively prevents MAL from growing too large in layers 9–17, where LAMB struggles to maintain control.
>
> As a result, our proposed optimizer can effectively limit the spike of MAL better in language models' large-batch training from both perspectives of theoretical analysis and empirical results.
>
> **Q2: Justification on scaling to large batch size.**
>
> **A2:** Thanks for your suggestion. We have made extensive experiments to provide a comprehensive comparison between AdamW, LAMB and MERIT. Here are the corresponding results.
>
> GPT2-Small with 2B training tokens:
>
> Validation loss:
>
> | Optimizer | bsz = 128 | bsz = 256 | bsz = 512 | bsz = 1K |
> |:-:|:-:|:-:|:-:|:-:|
> | AdamW| 3.172| 3.194| 3.288| 3.470|
> | LAMB | 3.189| 3.176| 3.247| 3.355|
> | MERIT| 3.205| 3.182| 3.214  | 3.280|
>
> LAMBADA Perplexity:
>
> | Optimizer | bsz = 128 | bsz = 256 | bsz = 512 | bsz = 1K |
> |:-:|:-:|:-:|:-:|:-:|
> | AdamW| 109.556| 100.836| 163.994| 279.166|
> | LAMB| 110.658| 96.300| 134.550| 169.678|
> | MERIT| 111.320| 93.789|113.665|154.482|
>
> GPT2-Medium with 8B training tokens:
>
> Validation loss:
>
> | Optimizer | bsz = 512 | bsz = 1K | bsz = 2K | bsz = 4K |
> |:-:|:-:|:-:|:-:|:-:|
> |AdamW| 2.830|2.871| 2.976| 3.172|
> |LAMB| 2.870|2.864| 2.919| 3.068|
> |MERIT| 2.841|2.853| 2.890| 2.982|
>
> LAMBADA Perplexity:
>
> | Optimizer | bsz = 512 | bsz = 1K | bsz = 2K | bsz = 4K |
> |:-:|:-:|:-:|:-:|:-:|
> | AdamW | 32.304 | 33.290| 57.657| 148.211|
> | LAMB | 45.900 | 38.484 | 52.780| 93.699 |
> | MERIT  |32.468 | 34.608| 39.174| 64.160 |
>
> **Analysis:**
>
> - **AdamW** performs well in small-batch settings but degrades rapidly as batch size increases, both in validation loss and LAMBADA perplexity.
> - **LAMB** stabilizes training at larger batch sizes but suffers from degraded generalization, especially visible in perplexity on LAMBADA.
> - **MERIT** consistently achieves lower validation loss and better perplexity across batch sizes, particularly in large-batch settings (e.g., bsz ≥ 2K).
> - The advantage of MERIT becomes more significant as model and batch size scale, demonstrating its effectiveness and robustness in large-batch optimization.
>
> **Q3: The ablation study on the effect of the norm choice is missing.**
>
> **A3:** Thanks for the advice. We have added an ablation study where we replace the weight-wise $l_2$-norm with the max-norm to compute the trust ratio, without using element-wise ratio and clipping, referred to as *maxLAMB*. The results are summarized below:
>
> **GPT-2 Small:**
> | Optimizer | Validation Loss |
> |:-:|:-:|
> |AdamW|3.470|
> |LAMB|3.355 |
> |maxLAMB| 3.304|
> |MERIT| **3.280**|
>
> **GPT-2 Medium:**
> | Optimizer | Validation Loss |
> |:-:|:-:|
> | AdamW | 3.172 |
> | LAMB | 3.068 |
> | maxLAMB | 3.040 |
> | MERIT | **2.982** |
>
> **Analysis:** The results show that simply replacing the $l_2$-norm with max-norm (maxLAMB) provides marginal gains over LAMB, but it still falls short of the performance achieved by MERIT.
>
> Additionally, as shown in Appendix K, we plot the trigger ratio for the lower bound of the weight-wise trust ratio during GPT-2 Medium training. The result reveals that the element-wise ratio and clipping play a crucial role in the early training phase, supporting the design choice behind MERIT.
>
> **Conclusion:**  The ablation confirms that MERIT's improvement does not stem from max-norm alone. Instead, its combination between element-wise trust ratio and clipping is key to achieving better performance.
>
> **Q4: it would be more clear to present the ablation results seperately in a table, rather than a figure of grid.**
>
> **A4:** Thanks for the advice. We will make corresponding revisions to present the ablation study separately in the updated paper.

---

> > ### Comment · Reviewer_QXND · 2025-04-06
> >
> > Thanks for the response. The numerical results shows that MERIT indeed improves large batch convergence over LAMB, though the improvement is a bit marginal.
> >
> > However, I still have a major concern on why MERIT's algorithmic design leads to reduced quantity of maximum logits. The small update magnitude does not imply the eventual small weight norm; it only establishes the bound for one update step (and it is just the upper bound). Hence, the argument in authors' response is not valid. Given that the main motivation of the algorithmic design is to reduce the maximum logits' quantity, justifying this point is critical.

---

> > > ### Author Response · Authors · 2025-04-06
> > >
> > > We sincerely thank the reviewer QXND for the further feedback. For the concerns and questions, here are our responses:
> > >
> > > **Q1: The improvement of MERIT over LAMB is a bit marginal.**
> > >
> > > **A1:** Thanks for your comment. We would like to emphasize that the performance gains achieved by MERIT are substantial rather than marginal.
> > >
> > > **Analysis:**
> > > - MERIT demonstrates superior scalability compared to LAMB across varying batch sizes. As shown in the results of my previous response, the performance gap between MERIT and LAMB widens with larger batch sizes.
> > > - MERIT also scales better across model sizes. Figure 4 illustrates that while the advantage of LAMB over AdamW diminishes as the model size increases, MERIT consistently outperforms LAMB. Notably, for GPT-2 Large, MERIT yields greater improvements over LAMB than LAMB does over AdamW.
> > >
> > > **Conclusion:** MERIT exhibits stronger scalability and more significant performance gains than LAMB across both batch and model scales.
> > >
> > > **Q2: Why MERIT's algorithmic design leads to reduced quantity of maximum logits.**
> > >
> > > **A2:** Thanks for your comment. We have provided a more detailed theoretical analysis of why MERIT is efficient in limiting the increase of max norm of query/key weights, and thus limiting the increase of max attention logit:
> > >
> > > **The influence of MERIT on upper/lower bound of weight max norm:**
> > >
> > > Given the query weight $W \in m \times n$ and updating steps of MERIT:
> > >
> > > $$
> > > W_{t+1} = W_t - \eta_t \cdot \frac{||W_t||_m}{||U_t||_m} \cdot U_t.
> > > $$
> > >
> > > Then,
> > >
> > > $$
> > > ||W_{t+1}||_m \leq ||W_t - \eta_t \cdot \frac{||W_t||_m}{||U_t||_m} \cdot U_t||_m \leq ||W_t||_m + \eta_t \cdot \frac{||W_t||_m}{||U_t||_m} \cdot ||U_t||_m = ||W_t||_m + \eta_t \cdot ||W_t||_m.
> > > $$
> > >
> > > Then we have:
> > >
> > > $$ ||W_{t+1}||_m \leq ||W_0||_m + \sum _{i=0}^t \eta_i \cdot ||W_i||_m $$
> > >
> > > Because $W_0$ is initialized with standard normal distribution, which means $ ||W_0||_m \approx \sqrt{2log(mn)}$:
> > >
> > > $$ ||W_{t+1}||_m \leq \sqrt{2log(mn)} + \sum _{i=0}^t \eta_i \cdot ||W_i||_m $$
> > >
> > > As for LAMB, we can have:
> > >
> > > $$ ||W_{t+1}||_m \leq \sqrt{2log(mn)} + \sum _{i=0}^t \eta_i \cdot ||W_i||_2 $$
> > >
> > > Due to the same usage of warm-up and cosine decay schedule of $\eta$ and the max $\eta$ of MERIT is smaller than LAMB in our experiments, and $\sum _{i=0}^t ||W_i||_m \leq \sum _{i=0}^t ||W_i||_2$, the upper bound of MERIT for  weights at each timestep is tighter than LAMB.
> > >
> > > Similarly, the lower bound of each updating step of MERIT is:
> > >
> > > $$ ||W_{t+1}||_m \geq ||W_t||_m - \eta_t \cdot \frac{||W_t||_m}{||U_t||_m} \cdot ||U_t||_m = ||W_t||_m - \eta_t \cdot ||W_t||_m $$
> > >
> > > Thus, the lower bound of weights at each step of MERIT is:
> > >
> > > $ ||W_{t+1}||_m \geq \sqrt{2log(mn)} - \sum _{i=0}^t \eta_i \cdot ||W_i||_m $.
> > >
> > > the lower bound of weights at each step of LAMB is:
> > >
> > > $ ||W_{t+1}||_m \geq \sqrt{2log(mn)} - \sum _{i=0}^t \eta_i \cdot ||W_i||_2 $.
> > >
> > > which is lower than MERIT.
> > >
> > > **Influence of Max Norm On Max Attention Logit:**
> > >
> > > We also discuss the influence of max norm of query and key metrices on final max attention logit:
> > >
> > > Given:
> > >
> > > $$
> > > \||W_Q\||_m = M_Q \quad \text{and} \quad \||W_K\||_m = M_K
> > > $$
> > >
> > > Each element in $ W_Q $ and $ W_K $ satisfies:
> > >
> > > $$
> > > 0 \leq |W_{Q,i,j}| \leq M_Q \quad \text{and} \quad 0 \leq |W_{K,i,j}| \leq M_K
> > > $$
> > >
> > > Each element in $Q$ and $K$ can be bounded as:
> > >
> > > $$
> > > Q_{i,k} = \sum_{m=1}^{n} X_{i,m} W_{Q,m,k}
> > > \leq \sum_{m=1}^{n} |X_{i,m}| M_Q
> > > = M_Q \sum_{m=1}^{n} |X_{i,m}|
> > > = M_Q \cdot C_X
> > > $$
> > >
> > > Similarly,
> > >
> > > $$
> > > K_{j,k} \leq M_K \cdot C_X
> > > $$
> > >
> > > where $C_X = \sum_{m=1}^{d} |X_{i,m}| $ is a constant representing the sum of absolute values in the input embeddings for a token and $d$ is the hidden size.
> > >
> > > The attention logit is:
> > >
> > > $$
> > > Logit_{i,j} = \sum_{k=1}^{d} \frac{Q_{i,k} K_{j,k}}{\sqrt{d}}
> > > \leq \sum_{k=1}^{d} \frac{(M_Q \cdot C_X)(M_K \cdot C_X)}{\sqrt{d}}
> > > = \sqrt{d} \cdot M_Q M_K C_X^2
> > > $$
> > >
> > > Therefore,
> > >
> > > $$
> > > Logit_{i,j} \leq \sqrt{d} \cdot M_Q M_K C_X^2
> > > $$
> > >
> > > Moreover, in decoder-only model implementations, each layer includes LayerNorm, which normalizes $X$ to follow a standardized normal distribution. This normalization effectively establishes upper bounds $C_X$.
> > >
> > > Thus,
> > >
> > > $$
> > > \text{Logit}_{i,j} \leq \sqrt{d} \cdot M_Q M_K C_X^2
> > > = \mathcal{O}(d ^ {2.5} \cdot M_Q M_K)
> > > $$
> > >
> > >
> > > This proof demonstrates that controlling the max norm of $ W_Q $ and $ W_K $ at each step effectively constrains the upper bound of the max attention logit because $d$ is a pre-defined parameter, which is crucial for large-batch training stability.
> > >
> > >
> > > **Conclusion:** MERIT’s max norm constraint effectively suppresses spikes in the max attention logit by enforcing both tighter lower and upper bounds on the weight norms at each optimization step. Experimental results from https://anonymous.4open.science/r/MAL-4K-2K-277A/ further demonstrate that MERIT consistently limits the max attention logits more effectively than LAMB.

---

### Official Review · Reviewer_ajhY · 2025-03-12

**Overall Recommendation:** 4

**Summary:**

The authors propose an extension of LAMB, which is an optimizer for large batch training. They propose to use an element-wise trust ratio using max norm and an element-wise clipping, rather than just an l2-norm based trust ratio as is the case for LAMB. The intuition is that the l2 norm is a poor surrogate for max norm, which is directly correlated with the max attention logit. Using max norm enables larger updates for more extreme weight values, thus prevening exploding logit spikes.

**Claims And Evidence:**

The main claim is that the max attention logits are the main difficulty when doing large batch training and that LAMB poorly addresses this by using a surrogate l2-norm based scaling. They provide evidence by showing that MERIT does reduce the max attention logits qualitatively in later layers and complimentary quantitive results showing good performance with very large batch sizes. The qualitative results are not too thorough in confirming that this does hold in practice, however, the theoretical results are sufficient.

**Essential References Not Discussed:**

None that I am aware of.

**Experimental Designs Or Analyses:**

They appear to be sound and valid.

**Methods And Evaluation Criteria:**

The authors conduct experiments training on the WikiText dataset with modestly sized GPT models and show good/comparable zero-shot perfance to AdamW despite using a much larger batch size. This seems to be sufficient, although there may be other large batch training methods > 2019 (LAMB) that need to compared with and that I am not too familiar with. Other reviewers might have some comments here.

**Other Comments Or Suggestions:**

For the readers less familiar with this topic, the authors should briefly describe the "trust ratio" earlier, or even in the abstract the conceptual idea. It will make those not very closely following this field more easily follow the introduction and motivation before going into the method.

**Other Strengths And Weaknesses:**

No other strengths or weaknesses.

**Questions For Authors:**

I am not finding it too clear why the it is called an "element-wise" trust ratio? From my understanding, the ratio s_{i,j} is the maximum along the ith row and jth column (of the weight matrix normalised by the updates). Can a different term be used to describe this modification? or can the authors more clearly explain why this naming is used - since the computation is not just dependant on w_{i,j} and u_{i,j}, but also the rows and columns w_{i, :}, w_{:, j).. etc.

**Relation To Broader Scientific Literature:**

Optimizers and large batch training is a very important topic for scaling training of moderately sized models.

**Theoretical Claims:**

The authors provide some more concrete derivations connecting the learning rate and attention logit values. This is important since MERIT scales the learning rate in such a way to avoid spiking the attention logits and subsequently improve training convergence and generalisation.

Additionally, the authors provide convergence derivations, which looks correct to me.

---

> ### Author Rebuttal · Authors · 2025-03-31
>
> We sincerely thank the reviewer ajhY for the valuable questions and comments. For the concerns and questions, here are our responses:
>
> **Q1: There may be other large batch training methods > 2019.**
>
> **A1:** Thank you for the comment. Most existing large-batch training studies focus on computer vision tasks (e.g., [1], [2], [3]), while discussions specific to autoregressive language models like GPT-2 or LLaMA remain limited.
>
> Although some works have attempted to improve LAMB's efficiency (e.g., [4]), they do not show performance gains. Up to now, LAMB remains the most widely used optimizer for large-batch training in language models.
>
> [1] Large Batch Optimization for Deep Learning Using New Complete Layer-Wise Adaptive Rate Scaling.
>
> [2] Achieving small-batch accuracy with large-batch scalability via Hessian-aware learning rate adjustment.
>
> [3] Revisiting LARS for Large Batch Training Generalization of Neural Networks
>
> [4] SLAMB: Accelerated Large Batch Training with Sparse Communication.
>
> **Q2: More qualitative plots showing the max attention logits across all layers.**
>
> **A2:** Thanks for your suggestion. We have drawn the max attention logits (MAL) across all layers in this anonymous link: https://anonymous.4open.science/r/MAL-4K-2K-277A/.
>
> **Analysis:**
>
> - In layers 1-8, where AdamW shows high MAL, MERIT and LAMB effectively suppress it.
> - In layers 9-17, MERIT reduces LAMB's MAL spikes via max-norm control.
> - In deeper layers (18-24), MERIT shows similar or slightly higher MAL, but the values are much smaller than in mid-layers and have minimal impact on convergence.
>
> **Conclusion:** The qualitative plots confirm that MERIT consistently mitigates extreme max attention logits spikes in critical mid-depth layers, where other optimizers struggle. This contributes to more stable training and improved convergence, particularly in large-batch settings.
>
> **Q3: The authors should briefly describe the "trust ratio" earlier.**
>
> **A3:** Thanks for your advice. We will put the introduction of the concept "trust ratio" earlier for clearer transmission in our revised paper.
>
> **Q4: Why the it is called an "element-wise" trust ratio?**
>
> **A4:** Thank you for your question. We initially referred to our proposed trust ratio as "element-wise" to distinguish it from the weight-wise trust ratio used in LAMB.
>
> As shown in Figure 3, the weight-wise trust ratio in LAMB can introduce inaccuracies since extreme values in one row or column may negatively affect the scaling of other rows or columns. To address this, we designed a finer-grained trust ratio that assigns different values to individual elements, reflecting variations in row/column distributions.
>
> To mitigate possible misunderstanding, we plan to revise it to **row and column-wise trust ratio** in the updated paper.
>
> **Additional Experiments:**
>
> We conducted additional experiments comparing MERIT and LAMB under the same settings in Table 1. The corresponding results are shown below:
>
> **GPT-2 Small**:
>
> | **Model**| **ARC ↑** | **COPA ↑** | **HellaSwag ↑** | **RACE ↑** | **WIC ↑** | **Avg ↑** |
> |-|-|-|-|-|-|-|
> | GPT-2 Small (AdamW-Batch Size=480)             | 43.43     | 66.00      | 29.20            | 29.00      | 50.16     | 43.56     |
> | GPT-2 Small (LAMB-Batch Size=4k)              | 44.40     | 64.00      | 28.46            | 27.46      | 50.16     | 42.90 |
> | GPT-2 Small (MERIT-Batch Size=4k)              | 45.83     | 67.00      | 28.82            | 27.56      | 50.16     | **43.87** |
>
> **GPT-2 Medium**:
>
> | **Model**                                      | **ARC ↑** | **COPA ↑** | **HellaSwag ↑** | **RACE ↑** | **WIC ↑** | **Avg ↑** |
> |-|-|-|-|-|-|-|
> | GPT-2 Medium (AdamW-Batch Size=480)            | 49.49     | 71.00      | 32.39            | 30.05      | 50.00     | 46.59     |
> | GPT-2 Medium (LAMB-Batch Size=6k)            | 47.47     | 72.00      | 30.90            | 28.52      | 50.31     | 45.84 |
> | GPT-2 Medium (MERIT-Batch Size=6k)            | 50.38     | 70.00      | 32.33            | 30.33      | 50.47     | **46.70** |
>
> **Analysis:**
>
> - LAMB's performance degrades on most tasks as batch size increases, confirming its sensitivity to scale.
> - MERIT consistently improves or maintains performance across tasks compared to AdamW in small-batch training, even with large batch sizes, demonstrating better generalization and stability.
> - On average, MERIT outperforms LAMB by ~1 point across downstream tasks, showing its practical advantage in both zero-shot and fine-tuning scenarios.
>
> **Conclusion:**
> These results reinforce MERIT's strength in scaling up batch size without compromising downstream performance. This further confirms that MERIT enables the use of larger batch sizes without sacrificing performance, in contrast to LAMB.

---

> > ### Comment · Reviewer_ajhY · 2025-04-04
> >
> > The authors have addressed all of my concerns. Adding the additional plots provided and also updating the introduction for the broader community will be great. I think this is quite important to have a widespread practical adoption in the community.
> >
> > I have read through the other reviews/rebuttal and I would be happy to see this submission accepted. I have updated my score accordingly.

---

> > > ### Author Response · Authors · 2025-04-04
> > >
> > > We sincerely thank reviewer ajhY for recognizing the value of our work and offering constructive feedback. We will incorporate the suggested changes in the revision. Thanks again for your time and effort in reviewing our submission.

---

### Official Review · Reviewer_1Wwm · 2025-03-16

**Overall Recommendation:** 4

**Summary:**

This paper proposes a new optimizer MERIT specialized for neural networks with self attention, especially Transformers, in a setting of large batch size training that typically lead low generalization performance. Based on observations that the training instability of Transformer comes from large amount of max norm at self attention weights, this method introduces (1) weight matrix-wise normalization (trust ratio) using max norm, (2) element-wise normalization with approximating with row/column vectors for efficiency, and (3) additional clipping of weight updates. Experiments show its rapidness of convergence on GPT-2 models compared with other optimizers, and yields better downstream performance.

EDIT after rebuttal: the authors answered my questions appropriately despite its laege burden of additional trainIng. Althouth the claim of the paper is limited for training flagship models, I think it is worth sharing the result to the community. I would raise my score.

**Claims And Evidence:**

The claims are straightforward to understand. The proposed method is based on an existing optimizer Adam and LAMB, but introduces more convervative method of regularization that are targeted on a pitfalls that the existing method overlooks.

**Essential References Not Discussed:**

NA

**Experimental Designs Or Analyses:**

The experiments are conducted on relatively small settings of experiments: GPT-2 with less than 1B params, batch with up to 8M tokens, trained with several billions of training tokens. These settings are significantly smaller than the current main-stream "small" pretrained models (e.g., Llama-3 8B, except the batch size: 8M tokens seems a common settings of training current models). I understand that there is a budget limitation to conduct experiments, but it seems results can't be generalized to the main-stream settings of LLM training.

**Methods And Evaluation Criteria:**

This is a paper proposing a drop-in replacement of LLM optimizer. So the outline of the method itself is not special.
Evaluation was conducted mainly on comparison of tendency during traiing LLMs, comparison of downstream results, and observation of inner weights and magnitude of activations. All of them supports advantage of the proposed method.
The experiments was conducted only by comparing existing optimizers, and any other methods than optimizers that tends to be adopted before replacing the optimizer, e.g., normalization techniques mentioned in 2.2, are ignored.

**Other Comments Or Suggestions:**

NA

**Other Strengths And Weaknesses:**

NA

**Questions For Authors:**

(1) tendency of long-run training: current LLMs are typically trained on trillions of tokens regardless of Chinchilla's law and I would like to understand what can be said if the proposed method was applied to such situation.
(2) interaction with treatments out of optimizer: In the real scenario, replacing optimizer is kind of the last choice due to its impact against the whole model, and other tweaks on the forward pass (e.g., reparametrization and normalization) tend to be applied to suppress instability. I'm also interested in how the proposed method works together with these methods, and whether applying the proposed method (more generally, replacing the optimizer from AdamW) is more important than other tweaks or not.

**Relation To Broader Scientific Literature:**

If the claim is generalized, it impacts against training schemes of LLMs, that may result reduction of huge amount of training cost.

**Theoretical Claims:**

The paper involves a proof of convergence of the proposed method. As detailed breakdown of the proof is moved to the appendix, I don't judge that the claim is correct. In an empirical perspective, the algorithm 1 seems to converge if Adam converges, as it involves only the components from Adam and additional normalization that suppress updates.

---

> ### Author Rebuttal · Authors · 2025-03-31
>
> We sincerely thank the reviewer 1Wwm for the valuable questions and comments. For the concerns and questions, here are our responses:
>
> **Q1: Tendency of long-run training.**
>
> **A1:** Thank you for the comment. Due to computational resource constraints, we are currently unable to apply our proposed optimizer to training runs involving trillions of tokens. As an alternative, we evaluate MERIT using GPT-2 models, which reflect architectural patterns commonly found in modern LLMs, to investigate its scaling behavior.
>
> The results below show that in large-batch settings, MERIT yields consistent improvements compared with LAMB as model size grows:
>
> | Optimizer | GPT2-Small | GPT2-Medium | GPT2-Large |
> |:---------------:|:----------:|:-----------:|:----------:|
> | AdamW           | 3.470      | 3.172       | 3.039      |
> | LAMB            | 3.355      | 3.068       | 2.971      |
> | MERIT           | **3.280**  | **2.982**   | **2.897**  |
>
> We observe consistent gains across all GPT-2 sizes under the Chinchilla training setting. As the model size changes, the gap between AdamW and LAMB decreases, while the performance gains of MERIT over LAMB remain stable.
>
> To further validate these findings, we conducted additional experiments using the open-sourced Llama implementation (https://github.com/kyleliang919/C-Optim). The results are as follows:
>
> | Optimizer | Llama-100M | Llama-250M | Llama-350M |
> |:---------------:|:----------:|:-----------:|:----------:|
> | AdamW           | 3.277      | 3.121       | 3.014      |
> | LAMB            | 3.257      | 3.096       | 3.001      |
> | MERIT           | **3.199**  | **3.046**   | **2.957**  |
>
> These results further support MERIT's scalability and its effectiveness in large-batch training across widely used LLM architectures.
>
> **Q2: Interaction with treatments out of optimizer**
>
> **A2:** Thank you for the comment. We investigated the effect of QK normalization in the large-batch training of GPT-2 small using 2B tokens. We first applied it to the AdamW optimizer:
>
> | Method | 8e-4 | 1e-3 | 2e-3 | 4e-3 | 6e-3 | 8e-3 | 1e-2 |
> |:-------------:|:----:|:----:|:----:|:----:|:----:|:----:|:----:|
> | AdamW + QK Norm | 3.608 | 3.593 | 3.524 | 3.475 | 3.527 | 3.496 | 3.671 |
>
> Compared to the AdamW baseline (loss = 3.470), QK normalization enables the use of larger learning rates without divergence. However, it consistently results in degraded performance in large-batch training scenarios, suggesting that its benefits in stability do not translate to better optimization outcomes in this setting.
>
> We then evaluated QK normalization in combination with the MERIT optimizer. Interestingly, MERIT not only maintains stability but also benefits from QK normalization:
>
> | Model       | MERIT (lr = 9e-3) | MERIT + QK Norm (lr = 1e-2) |
> |:-----------:|:-----------------:|:----------------------------:|
> | GPT2-Small | 3.280             | **3.240**                    |
>
> **Analysis:** The introduction of QK Norm on MERIT successfully reduces the validation loss further, and the potential reason is that the use of element-wise trust ratio enables more stable updating when integrated with QK Norm.
>
> To ensure a fair comparison, we use a vanilla implementation of GPT-2 (nanoGPT) throughout our experiments when evaluating the proposed optimizer. We will consider more combinations of MERIT with other tweaks in the future!
>
> **Conclusion:** While QK normalization alone does not improve performance with AdamW in large-batch training, it complements MERIT effectively, leading to further performance gains. This demonstrates MERIT's flexibility and its potential to integrate well with auxiliary techniques for scaling up training.

---

### Decision · Program_Chairs · 2025-05-01

**Decision:**

Accept (poster)

**Comment:**

In this paper, the authors propose a modification to the LAMB optimizer by replacing the L2 norm with the $l_\infty$ norm for trust ratio estimation. They provide a theoretical convergence analysis to support this modification. However, I agree with reviewer QXND to some extent that the argument concerning controlling the maximum attention logits appears limited and somewhat trivial. The paper would significantly benefit from a more comprehensive analysis of the impact of different update scales, and why this benefits the large batch training, as suggested by the reviewer.

Nonetheless, this discovery remains noteworthy for the community. It is also interesting to see that selecting different norms can influence specific training dynamics properties.